# Palaeodemographic modelling supports a population bottleneck during the Pleistocene-Holocene transition in Iberia

Javier Fernández-López de Pablo [1,2], Mario Gutiérrez-Roig [3], Madalena Gómez-Puche[1,2],
Rowan McLaughlin [1,2], Fabio Silva[4] & Sergi Lozano [1,2]

Demographic change lies at the core of debates on genetic inheritance and resilience to climate change of prehistoric hunter-gatherers. Here we analyze the radiocarbon record of Iberia to reconstruct long-term changes in population levels and test different models of demographic growth during the Last Glacial-Interglacial transition. Our best fitting demographic model is composed of three phases. First, we document a regime of exponential population increase during the Late Glacial warming period (c.16.6-12.9 kya). Second, we identify a phase of sustained population contraction and stagnation, beginning with the cold episode of the Younger Dryas and continuing through the first half of the Early Holocene (12.9-10.2 kya). Finally, we report a third phase of density-dependent logistic growth (10.2-8 kya), with rapid population increase followed by stabilization. Our results support a population bottleneck hypothesis during the Last Glacial-Interglacial transition, providing a demographic context to interpret major shifts of prehistoric genetic groups in south-west Europe.

[1] Institut Català de Paleoecología Humana i Evolució Social (IPHES), Edificio W3, Campus Sescelades URV, Zona Educacional 4, 43007 Tarragona, Spain. [2] Àrea de Prehistòria, Universitat Rovira i Virgili, Avda. Catalunya 35, 43007 Tarragona, Spain. [3] Behavioural Science Group, Warwick Business School, University of Warwick, Scarman Rd, Coventry CV4 7AL, UK. [4] Department of Archaeology, Anthropology & Forensic Science, Faculty of Science & Technology, Bournemouth University, Fern Barrow, Poole, Dorset BH12 5BB, UK. Correspondence and requests for materials should be addressed to J.F.-L.d. P. (email: jfernandez@iphes.cat)

Demographic change lies at the core debates on genetic inheritance, cultural evolution and resilience to climate change of prehistoric hunter-gatherers[1–4]. The Final Pleistocene and the Early Holocene archaeological record provides large ensembles of radiocarbon dates which can be used as a proxy to infer long-term demographic dynamics. Under the assumption that past population levels were proportional to the amount of anthropogenic carbon accumulated and randomly dated, previous studies have demonstrated the suitability of using summed probability distributions (SPD) of calibrated archaeological radiocarbon dates to reconstruct relative changes in population size[5–7].

Although this method is prone to some biases due to sampling, taphonomic factors and the uncertainty of the calibration process, recent contributions have introduced specific protocols to overcome such problems[8–10], and have refined previous assumptions about the relationship between past population levels and the accumulation of anthropogenic carbon[11]. In particular, the application of stochastic simulations in the SPD analysis[8,9,12], has allowed noise produced by sampling error and the calibration process to be filtered out from the signal caused by actual changes in population densities over time.

Therefore, the radiocarbon record has become a central line of evidence for reconstructing population dynamics of prehistoric hunter-gatherers around the world. Following the SPD methodology, previous studies have shown a causal relationship between climatic deterioration and population contraction during the Last Glacial Maximum (LGM c.23-19 kya) in Europe[13,14], a finding independently validated through climate envelope modelling[15].

Recent studies in the field of SPD analysis[16] have evaluated different models of demographic growth for prehistoric hunter-gatherers during the Pleistocene–Holocene transition[17,18]. In Wyoming-Colorado (North America), it has been proposed a long-sustained regime of exponential demographic increase between 13-6 kya, with annual growth rates around 0.04%[17]. In South America, Goldberg et al.[18], found a regime of logistic demographic growth between 14-5.5 kya, with very high initial growth rates of 0.13% between 14-9.5 kya, followed by a long period, between 9.5–5.5 kya, of stagnation in population size. However, while these case studies are illustrative of rapid demographic growth following initial colonization, and ultimately inhibited by carrying capacity, it is not clear that similar demographic trajectories took place in other parts of the world during the Pleistocene–Holocene transition, especially in areas with earlier human presence and more prolonged socio-ecological interactions.

In Europe, emerging ancient DNA evidence suggests that Late Glacial and Early Holocene population dynamics were associated with a major genetic substitution. This event saw the so-called Mirón genetic cluster dated c.19-15 kya and related to the Magdalenian culture replaced by the Villabruna genetic cluster at 14-7 kya[19]. However, the demographic context of such a genetic shift remains to be investigated and only the archaeological radiocarbon record provides the necessary resolution to reconstruct changes in population levels at regional scale. The Iberian Peninsula, because of its cul-de-sac position in the southwestern extreme of the Eurasian continent, remains central to discussions about demographic processes involving human refuges, migrations, population admixture and replacements. In as far as the Last Glacial-Early Holocene transition is concerned, Iberia provides a unique opportunity for reconciling population histories drawn from ancient DNA (aDNA) and radiocarbon paleodemographic modelling, a major methodological challenge in population paleostudies[20].

In Iberia, few previous works have examined the relationship between demography and climatic-environmental change during the Late Glacial and the Early Holocene using SPD analyses from different geographic scales. From an European perspective, Gamble and colleagues[13,14] proposed that Northern Iberia was a refugium for human populations during the LGM. This caused a local increase in population density in the Cantabrian and Aquitanian regions during the LGM, and subsequent Late Glacial recolonization of Western Europe was undertaken by the descendants of this population[14]. Subsequent research has examined the radiocarbon record of some Iberian regions such as the Mediterranean[21,22] and the Cantabrian strip[23]. However, because these previous studies were based on initial applications of the SPD technique, neither the effects of the calibration curve nor the statistical significance of the observed patterns were tested.

In this work we analyze empirical radiocarbon time series to reconstruct relative population changes and test several alternative demographic models during the Pleistocene–Holocene transition in Iberia (c.16.6-8 kya). We investigate patterns of regional variation in the observed trends and the relationship between the inferred demographic regimes with climatic change. Our results allow an independent validation of population histories drawn from aDNA data, thereby adding to the understanding of human population dynamics during a pivotal moment in Europe's Prehistory.

## Results

**Dataset**. We collected and analyzed a new database of 1198 radiocarbon determinations from 795 assemblages and 246 archaeological sites for the time range under analysis (Fig. 1 and Supplementary Methods). After thorough evaluation of the radiocarbon record and the application of filtering criteria (Supplementary Methods) we excluded from the analysis 291 dates and retained 907 determinations. The size and quality of this data set meets the conditions for a statistically reliable SPD[5,24]. The pruned data set has been calibrated using the atmospheric and, when necessary, the marine Intcal 13 calibration curve[25], applying different local reservoir corrections (Supplementary Methods, Supplementary Fig. 3 and Supplementary Table 5). In order to avoid edge effect problems, we have defined a broader time window of (18–7.5 kya). The results are presented in two different subsections: First, we model population dynamics in terms of relative population changes using SPD time-series and bootstrap simulations, and then we estimate different phases of demographic growth rates as derived from dynamic growth models.

**Relative population changes**. Figure 2 shows the results of the taphonomically corrected SPD and bootstrap simulations (Methods section and Supplementary Methods). The observed time series displays a long-term increasing trend with five main pulses (lasting between 3 and 6 centuries each) placed at 18–17.6, 14–13.8, 13.7–13.4, 13–12.7 and 7.8–7.6 kya.

Following recent studies[6,8,9], we tested the statistical significance of the relative population changes during the study period by fitting an exponential model of simulated radiocarbon dates for the whole time-range under analysis. The exponential model was chosen because it is the only simple function that provides an adequate null model of millennial-scale changes in population size. Of the alternatives, a uniform null model implies a regime of stationary growth arising from complex interactions of fertility and migration and is thus not appropriate as a null model for such a long chronological span. On the other hand, a logistic model implies a carrying capacity threshold, which in any case could change with time, especially in the context of the present

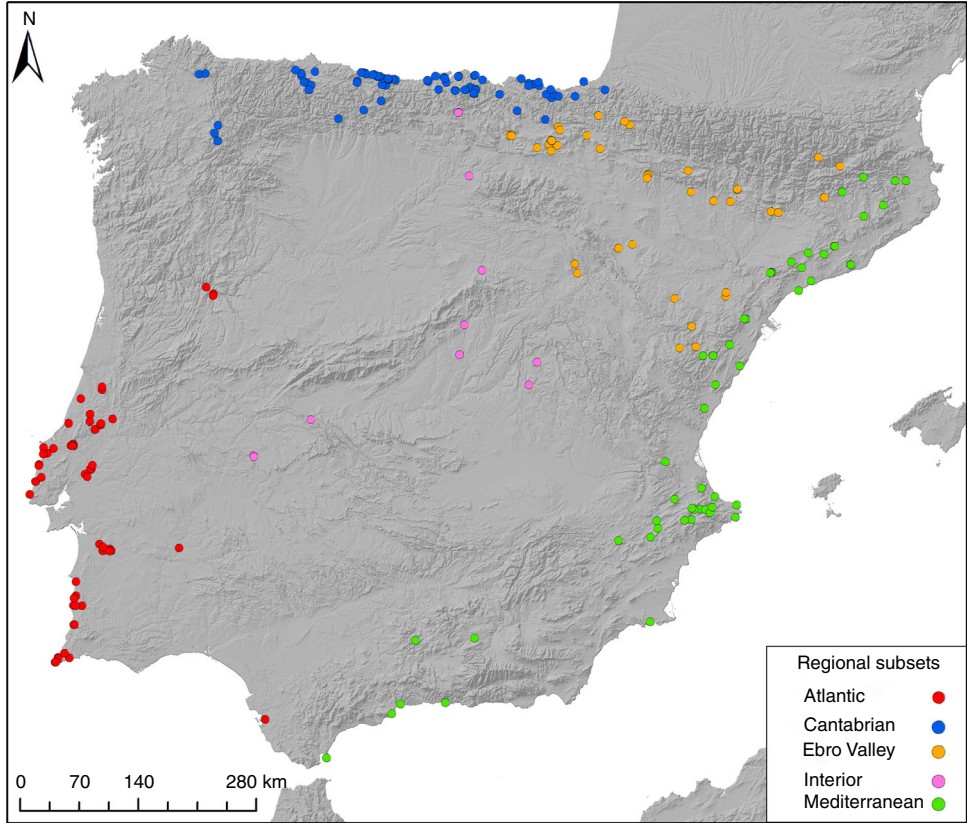

**Fig. 1** Distribution of archaeological sites with radiocarbon dates analyzed in this study. The colours represent the different regional subsets analyzed. The background map has been generated in ArcGIS 10.3 using 1 arc second SRTM raster data (available from https://earthdata.nasa.gov/) to represent terrain via a hillshade model

study, which is conducted over the dynamic environments of the Pleistocene–Holocene transition. Hence, the exponential null model was used to identify positive and negative deviations in the reconstructed population trends (see Methods and Supplementary Methods for more details). Figure 2d displays the population proxy plotted against the 95.4% CI envelope of the exponential null model of demographic growth. We found significant overall departures from the null exponential model (p value < 0.001).

The population proxy shows five periods of significant positive, and four others of significant negative deviations from the exponential null model. The first of these occurs at the beginning of the time series, during the pleniglacial (18–17.6 kya), coinciding with the initial stages of the Middle Magdalenian culture. From 17.6 kya, we observe an abrupt decline followed by gradual increase in relative population densities. These changes occurred during the harsh climatic conditions of the Heinrich stadial 1 and the rapid warming of the Last Greenland Interstadial commencement (GI-1e or Bølling chronozone).

A second period of population increase is clearly identified between c.14-12.7 kya, showing minor fluctuations with three peaks (at 14–13.8, 13.7–13-4 and 13–12.7 kya) above the exponential null model of demographic growth, roughly covering the Final Upper Magdalenian-Early Azilian/Epipaleolithic transition. After the third brief spike (c.13–12.7 kya) the population densities abruptly decline from c.12.7-12.4 kya. In as far as we can be sure, given the natural limits of radiocarbon resolution which here is compounded in uncertainty by the marine reservoir effect[26], this decline coincides with the onset of the climatic effects of the Younger Dryas (YD) in the western Mediterranean. From 12.4 to 9.9 kya, we observe a long period of stable population sizes roughly covering the Epipaleolithic and the onset

of the Early Mesolithic, manifesting in the data in three major negative deviations from the exponential null model at 11.7-11.3, 11.1-10.7 and 10.5–9.8 kya, covering most of the first half of the Early Holocene. This prolonged phase ended during the Early Mesolithic period, at c.9.8 kya, when the relative population densities abruptly increased to reach a positive pulse c.9.7–9.1 within the 95% CI envelope of the exponential null model. However, the high activity levels were not sustained; we observe next an abrupt decline in the population proxy, leading to a short period between 9 to 8.7 kya negatively departing from the null exponential model. Finally, we notice a pronounced recovery from 8.7 to 7.8 kya, during the Late Mesolithic. This last episode includes a narrow trough around 8 kya which might be correlated with the impact of the 8.2 kya cal BP cold event on Late Mesolithic populations[27,28]. At the Iberian scale, however, this climatic event did not produce any statistically significant change to relative population levels. Finally, we observe a narrow peak in the population proxy at c. 7.8–7.59 overpassing the CI interval envelope of the exponential null model to abruptly fall again at the end of the time series.

To investigate the patterns of regional variation in the inferred demographic dynamics, we divided the Iberian radiocarbon record into five regional subsets: Cantabrian, Ebro Valley, Atlantic, Mediterranean and Interior (Supplementary Figures 13 and 14). For each regional subset we produced both a SPD population proxy and a simulated null model of exponential growth by subsampling the Iberian radiocarbon record with the same number of bins contained in each regional empirical data set (Fig. 3). Each regional null model of exponential growth was expressed as a 95.4% CI envelope, allowing the identification of positive and

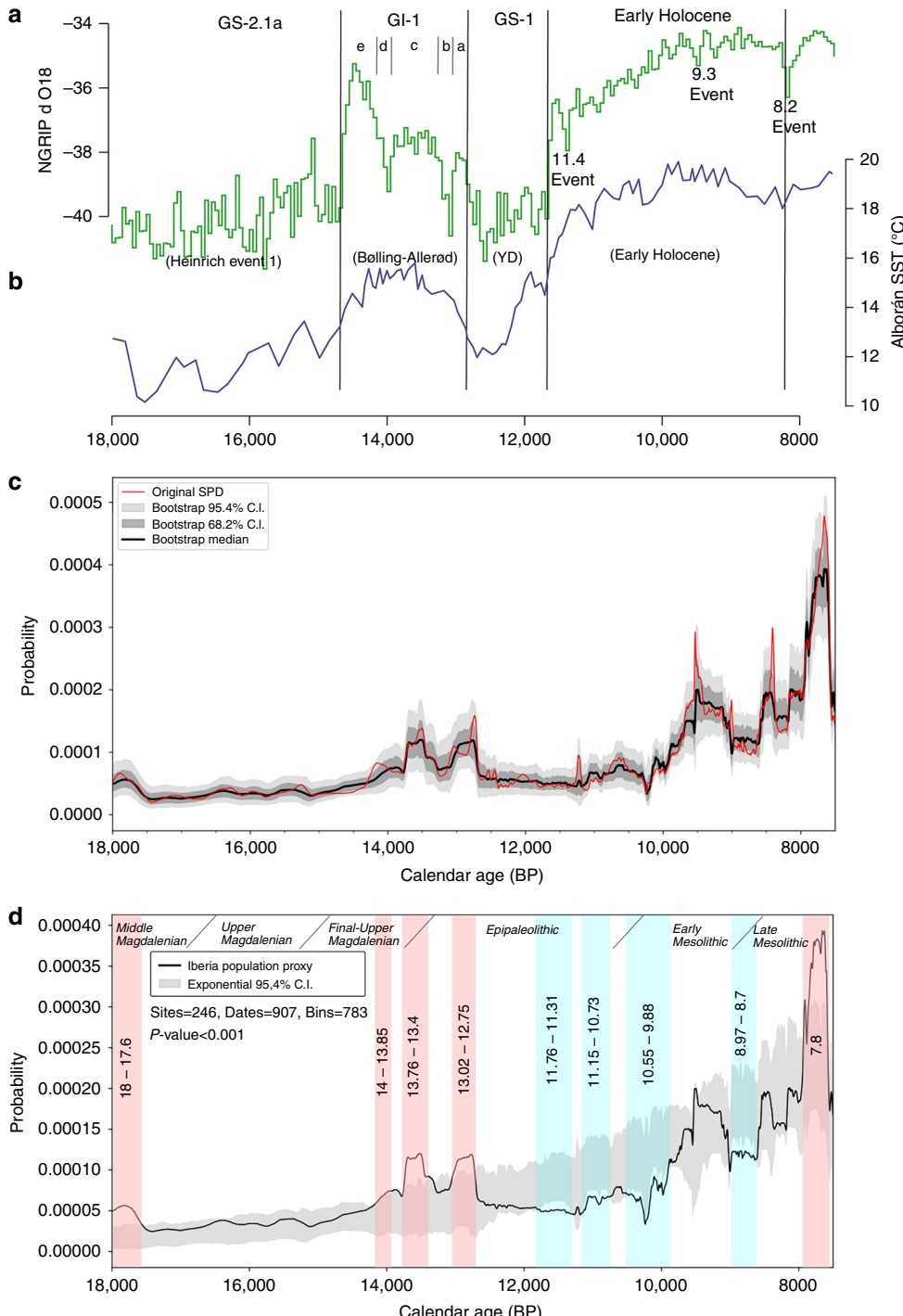

**Fig. 2** SPD inferred population density changes between 18,000 and 7500 cal BP for Iberia. **a**, **b** Palaeoclimatic framework of the Last Glacial-Interglacial transition. **a** The green curve represents the variation of [18]O from the NGRIP ice-core record with the Greenland stratotype chronology[77]. **b** The blue line depicts the reconstruction of Sea Surface Temperature based on the deep-sea core MD95-2043 in the Alborán sea, considered as a palaeoclimatic proxy specific to the Iberian Peninsula[56]. **c** Summed Probability Distribution (SPD) of the calibrated radiocarbon dates after applying bootstrap simulations on the taphonomically corrected SPD curve (in black) (seeMethods). **d** Population proxy (in red) plotted against a bootstrapped null model of exponential growth. Light red and light blue shaded regions denote positive and negative deviations from the null model of exponential demographic growth respectively. The global *p* value is calculated using the computational method of Timpson et al.[8] for a set of simulated SPDs and express the global significance of the curve's departure from the exponential null model

negative departures from the panregional model. Obviously, the division of the radiocarbon record into smaller regional or sub-regional subsets reduces the sample size and sampling density at the expense of statistical significance of the inferred trends. In turn, the smaller sample increases the

effect of non-demographic confounding factors such as research and sampling biases inherent to the observed trends (Supplementary Discussion). For these reasons, a conservative approach was taken to the interpretation of regional patterns.

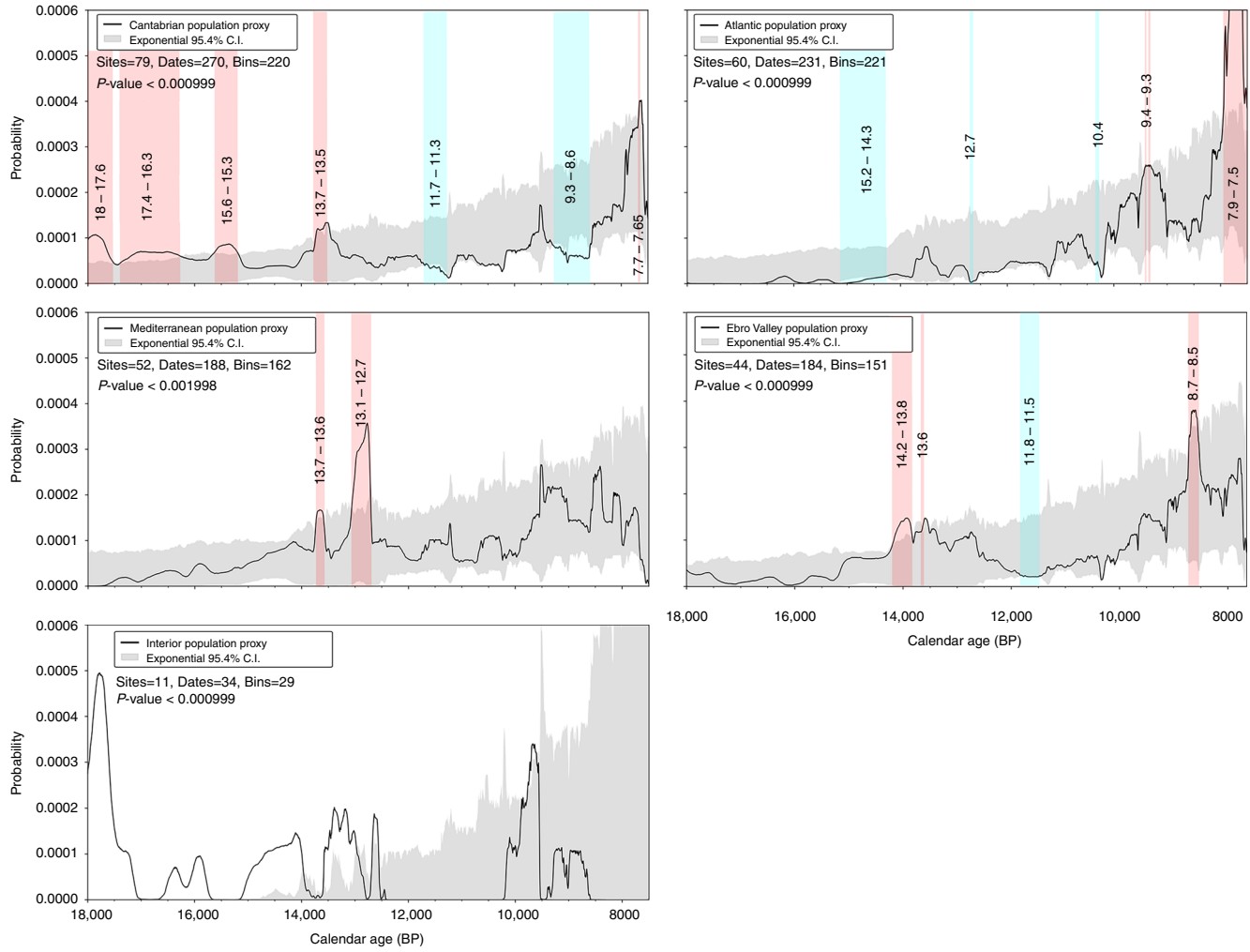

**Fig. 3** SPD inferred population density changes between 18,000 and 7500 cal BP for each regional subset. The black bold line represents the population proxy based on the median of the taphonomically corrected SPD after applying bootstrap resampling. For each regional subset is compared with a null model of exponential demographic growth (in gray). The vertical pink and blue bands represent periods of significantly higher and lower population densities respectively compared to the null exponential model. The global *p* values are calculated using the computational method of Timpson et al.[8] for a set of simulated SPDs and express the global significance of the curve's departure from the null model

Overall, the long-term changes identified at Iberian scale show a good agreement with the patterns identified in those regions with more representative data sets. We found significant differences (*p* value < 0.05) between the inferred population patterns and the null exponential model in the Cantabrian, Mediterranean, Atlantic and Ebro valley subsets. The Iberian Interior subset shows major gaps in the time series as a result of its lower sampling density[29]. This prevented us from obtaining any meaningful comparison between the SPD and the exponential null model.

Aside from the poorly powered Iberian interior, all regional subsets show very similar trends: the population significantly increased during the second half of the Last Glacial Interstadial, decreased during the Younger Dryas and the first half of the Early Holocene and increased again during the second half of the Early Holocene. The greatest contrast with the panregional model is observed in the Cantabrian subset, during a long period (c. 18-16.3 kya) when population levels were sustained well above the 95.4% CI of the exponential null model. This period is part of a broader episode of population increase previously reported in the Cantabrian region following the Last Glacial Maximum[14], and in the time series we analyzed, the increased population also spans the Middle

Magdalenian and the beginning of the Upper Magdalenian cultural periods.

Under the general pattern of population increase at the end of the Last Glacial period, it is worthy to note slight inter-regional differences in the turning point of population decline. For instance, in Cantabria and the Ebro valley subsets it is dated *c.* 13.5 kya during the final stages of GI-1, whereas in the Mediterranean region it is dated to *c.* 12.9-12.7 kya, during the onset of the YD. Such divergences suggest different population responses to regionally-specific conditions which require further investigation using multi-proxy archaeological and palaeoenvironmental evidence at sub-regional level. In the Cantabrian region, it seems plausible that the high population[30] reached a natural carrying capacity at this time, tilting the trend towards long-term decline. By contrast, the population of the Mediterranean region probably never reached such high levels, but was particularly exposed to challenges caused by the abrupt changes to climatic and environmental conditions associated with the YD[21].

A second difference between the panregional model and the regional population trajectories is observed at the end of the time series (c.8–7.5 kya) which seems sensitive of edge effects on the inferred regional population trajectories. During this short span the population proxy of the Cantabrian and Atlantic subsets show

a significant peak. By contrast, in the Mediterranean and Ebro valley subsets, population proxy remains within the confidence interval envelope of the exponential null model until an abrupt decay at 7.5 kya, at the end of the time series. As extensively discussed in the Supplementary Discussion, besides the edge effect problem, these inter-regional differences at the end of the time series are caused by a combination of different factors, namely: (i) intense research pressure upon shell midden archaeology in the Atlantic and Cantabrian regional subsets[31], which has produced a concentration of radiocarbon dates on marine shells and thus a pronounced spike in the SPD signal (Supplementary Figure 10), and (ii) the sharp decrease of Mesolithic activity synchronous with the arrival and spread of farming. Mesolithic populations persisted in the Mediterranean and Ebro valley subsets until c.7.6-7.3 kya[21], whereas in the Atlantic and Cantabrian regions, foraging systems continued to c. 7.2-7 kya and 6.2-5.9 kya respectively[32–34]. For these reasons, we have excluded all dates after 8,000 cal BP from the subsequent modelling exercises.

**Demographic growth rates**. The exponential fit estimates the mean annualized growth rate $0.01958\% \pm 3 \times 10^{-8}\%$ ($r = 0.80$) which corresponds to a seven-fold population increase during the time-range 18-8 kya. The radiocarbon derived time series provide evidence to reject the null hypothesis ($p < 0.000999$) that exponential growth was the sole explanatory framework for the population dynamic. Therefore, we have explored alternative, more complex hypotheses using an information-theoretic based model selection approach. Rather than opting for a full dynamic growth rate, as proposed by Brown[16], we opt for a more parsimonious approach that keeps the number of changes in demographic regime to a minimum, whilst remaining alive to two natural breakpoints that coincide with well-established climate and environmental transitions in Iberia. The first of these relates to the onset of the YD around 12.9 kya and the second with the onset of the Early Mesolithic around 10.2 kya. The explanatory power of six different models (Models A–F, Fig. 4) have been

compared against the empirical population proxy and ranked using Akaike and Bayesian Information Criteria (see[35], Supplementary Methods), for the period between the Upper Magdalenian and the end of the Late Mesolithic (16.6-8 kya). Since our research objectives are to definite long-term demographic patterns, we have required phases to last at least a 2000 years; this also avoids overfitting models to calibration curve artefacts, which occur over relatively short spans of time

The six selected models increase in complexity -two single-phase models (A and B), two two-phase models (C and B) and two three-phase models- and assume different hypotheses about demographic growth regimes for each phase and the transitions between phases. The two simplest ones (A and B) fit a simple parametric growth model -exponential and logistic- for the time range under analysis (c.16.6-8 kya). The exponential growth has been widely applied to study short-term demographic bursts of invasive species occupying new niches[36] as well as recent explosive human demographic growth[37]. In recent radiocarbon palaeodemographic research, exponential growth has been empirically demonstrated for long-term demographic dynamics of prehistoric hunter-gatherers[17], while in other studies it has been proposed as a null hypothesis to infer phases of statistically significant high or low population levels[6,8,9,38,39]. Under an exponential growth, our Model A assumes long-term continuous, gradual and uninhibited population increase from the beginning of the Upper Magdalenian to the end of the Late Mesolithic.

The second single-phase demographic model is the Model B, fitting a logistic function[40] in which the growth rate decreases as the density of population approaches carrying capacity, which is the population threshold that a given environment can support. This kind of density-dependent growth has been explored in different simulations of Late Pleistocene human dispersals[41,42] and empirically tested in more recent radiocarbon paleodemographic studies on both prehistoric hunter-gatherers[18] and agriculturalists[43].

Models C and D represent two different two-phase models, consisting on two consecutive growth regimes -exponential plus

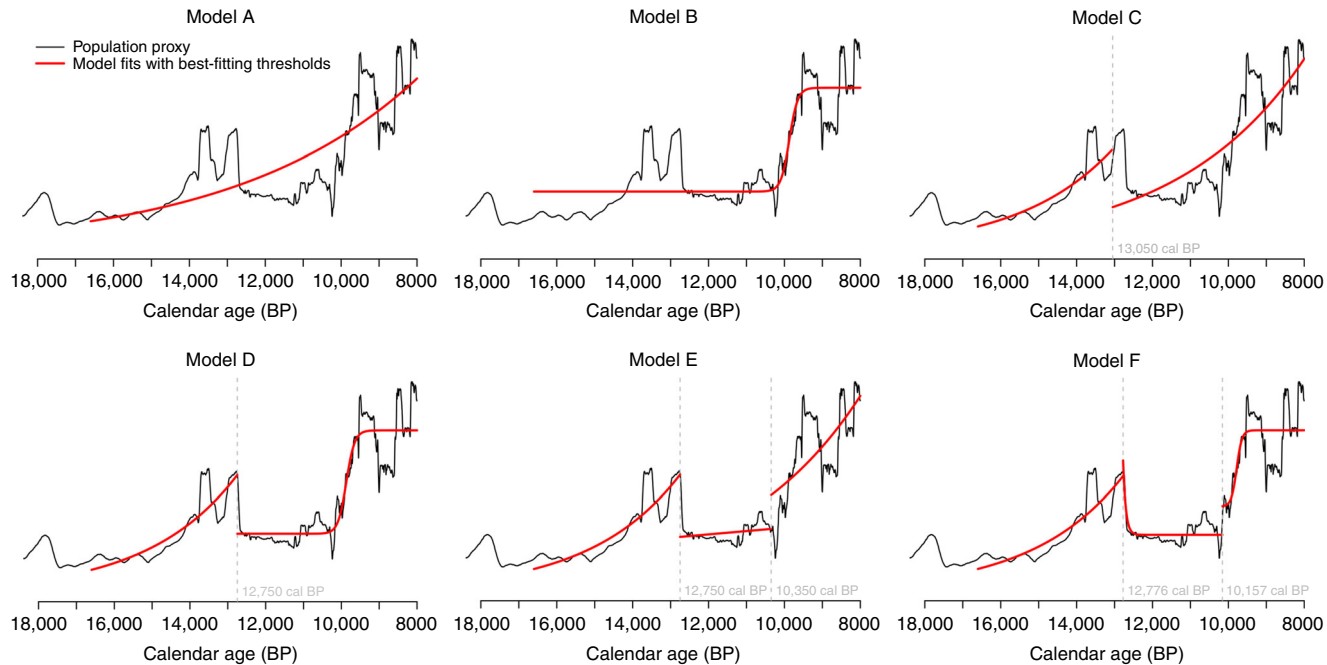

**Fig. 4** Demographic models tested using an information-theoretic based model selection approach. The black curve depicts the empirical SPD population proxy whereas the red curves represents the model fits with best fitting thresholds for each model. The vertical gray dashed lines represent the highest likelihood breakpoint values

**Table 1 Growth rates derived from the model E for each demographic phase in Iberia (18-8 kya)**

|  | Whole range | Phase 1 | Phase 2 | Phase 3 |
|---|---|---|---|---|
| Range (kya) | 18-8 | 16.6-12.78 | 12.78-10.12 | 10.12-8 |
| Mean Annual Growth Rate (%) | 0.01958 | 0.0411 | $a \exp(-b\,t) + c$, $a = 4.54 \times 10^{-24}$; $b = 0.00339$; $c = 0.0000555$ | $A/(1+\exp((x_0-t)/s)) + c$ $A = 0.000080$ $x_0 = 9799$ $s = 77.89$ $c = 0.000078$ |
| Min. 95.4% CI Max. 95.4% CI | 0.01958-0.01958 | 0.0402-0.0420 | $-0.7908 \approx 0$ | $\approx 0$ 0.2278 |
| Taxonomical units |  | Upper Magdalenian, Early Azilian, Epimagladenian, Epipaleolithic | Late Epimagdalenian, Late Azilian, Epipaleolithic, Sauveterrian | Early Mesolithic, Late Mesolithic |

The confidence intervals for the whole range and the different phases of the demographic dynamic model are also presented. The bottom row provides the correspondence of the demographic phases with the archaeological taxonomical units

exponential or exponential plus logistic- separated by a unique breakpoint around the YD commencement. This abrupt transition would signify a rapid and catastrophic decline of population levels associated to the onset of the YD, as simulated by Buchanan and colleagues'[44] in North America. On one hand, Model C assumes that there were similar processes of exponential demographic growth operating before and after the YD. On the other hand, the two consecutive demographic regimes assumed for the Model D -exponential before and logistic after the YD- predict an abrupt population increase for the period 10.2-9.5, and a sustained apogee spanning 9.5 to 8 kya, modeling when the maximum carrying capacity was reached.

Finally, models E and F exemplify two different versions of a three-phase model of population dynamics. Both models infer the same demographic processes for phases 1 and 2, while differing for phase 3. Thus, an exponential growth is fitted to the first phase, spanning the beginning of the Upper Magdalenian to the YD commencement (c.16.6-12.9 kya), whereas the second phase (12.9-10.2 kya) is modelled as an exponential decay towards a limiting value. Importantly, it should be noted this second phase contrasts with the classic parametric growth models (e.g., exponential and logistic) used for models A-D. Instead of assuming an abrupt and catastrophic decline of population associated with the YD (as proposed in models C and D), this phase 2 aims to model a more complex transition from negative to stationary (null) demographic growth, where migration, natural fertility and mortality rates were all held in balance. Such an assumption can be derived from other models or demographic hypotheses about processes operating during phases of lowered population levels or densities. One such model is that of strong Allee effects[45], which in the case for humans and primates foresees negative growth rates if populations fall below a critical density threshold[46]. Other relevant hypotheses are the stationary demographic growth that followed the boom and bust pattern identified after the Neolithic demographic transition;[6] or the foraging-resource disequilibrium model[47], which predicts stationary demographic growth under environmentally driven changes in carrying capacity. In order to explore different plausible scenarios for the third demographic phase (c. 10.2-8 kya), models E and F differ in how they model the underlying trend for the third demographic phase, assuming for the whole Mesolithic an exponential (model E) or logistic (model F) growth.

In the model fitting process, the values of the breakpoints between phases are important parameters as they influence the overall goodness-of-fit. For the results presented in this paper, we consider two already mentioned breakpoints, corresponding to the onset of the YD (12.9 kya) and the Early Mesolithic (10.2 kya). However, the timing of these transitions is not as sharp and well-dated as these often-cited dates seem to portray them, nor do they occur contemporaneously across the whole of the Iberian Peninsula. To account for this, we have allowed these breakpoints

to vary by one-year increments ranging 150 years before and after the above-mentioned dates. This provides a measure of uncertainty around our inferences, especially the growth rates, and accounts for the sub-regional variability in the timing of the transitions.

We found that single and double phase population trajectories (models A and B), whether exponential or logistic poorly fit with the empirical population proxy. Two-phase models (C and D), which accommodate a catastrophic population decline at the onset of the YD, are a much better fit for the data, and the three phase models (E and F) represent further refinements. The best fitting model of all was model F. The mean annual growth rates predicted by this model and their 95.4% CI are detailed in Table 1 and Fig. 5 (bottom). In this model, a distinctive demographic phase of exponential growth started with the Upper Magdalenian (c. 16.6 kya), (which covers different regional archaeological taxonomies, see Supplementary S7), and extends to the beginning of the Younger Dryas (c. 12.75 kya). During this phase, the annual growth rates derived from the local exponential fit range from 0.040% to 0.042%.

From the 12.75 kya and until 10.2 kya in the Early Holocene, a second demographic phase can be characterized by annual growth rates falling to negative values (initially as strong as −0.79%), followed by a recovery towards a long-sustained period stationary growth. Different archaeological taxonomic units have been identified during this phase according to different regional sequences and research traditions (Late Azilian, Epipaleolithic, Late Epimagdalenian and Sauveterrian).

Finally, a third demographic phase, dated between 10.2-8 kya, spanning the Early and the Late Mesolithic periods, can be characterized by logistic growth. Under this logistic model, we find a sub-phase of abrupt population increase at the beginning of the Early Mesolithic (c.10.2-9.8 kya) with mean annual growth rates rising from 0 to 0.22%. This predicts that the population approximately doubled every 12 generations. As discussed later, such a rate of population increase is higher than the endogenous logistic growth of other prehistoric hunter-gatherers reported by SPD studies[18], but it is important to note that this sub-phase was only sustained for a few centuries at most. From c. 9.8 kya, the logistic model predicts a second sub-phase of continuous decelerating growth rates to 0, followed by a period of relatively stable population levels from 9.3 to 8 kya.

## Discussion

Our results support a three-phase model of long-term population dynamics for the Last Glacial to the Early Holocene transition in Iberia (c. 16.6-8 kya). Both the inferred relative population changes and annual growth rates derived from the dynamic model have significant implications for discussing (i) the relationship between demography with climate-environmental change and subsistence patterns in Iberian prehistory (ii) the

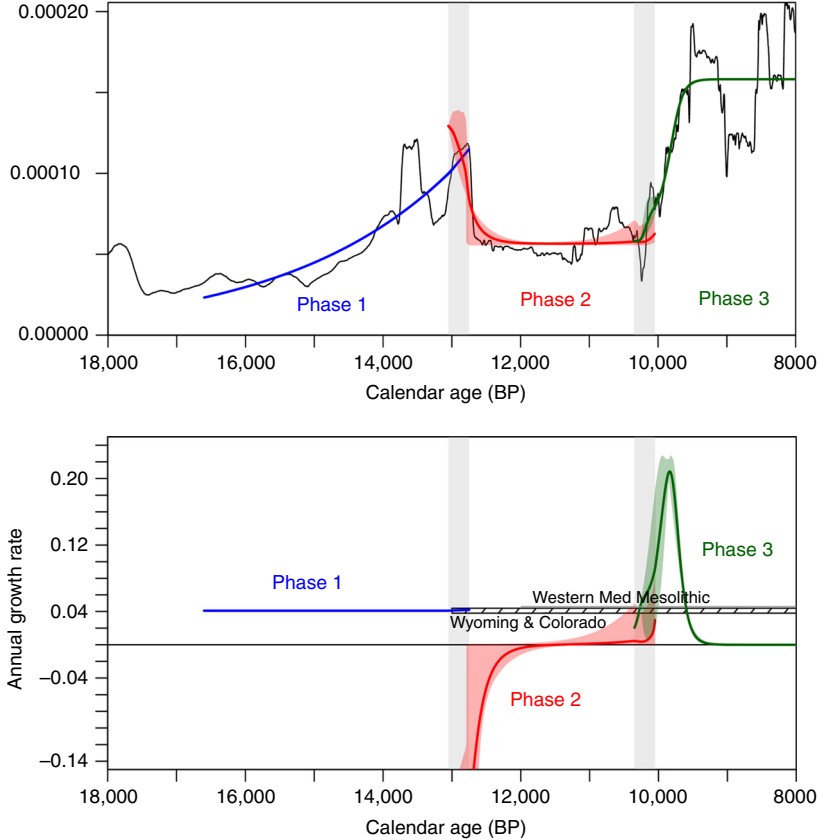

**Fig. 5** Regression fits and demographic growth rates of the best-fitting model. Top: Population proxy (in black) and regression fits of the best-fitting model of demographic dynamics. Bottom: Comparison of inferred annual growth rates derived from the best-fitting model with those derived from other case studies of prehistoric foragers in Wyoming & Colorado[2] and the Western Mediterranean Mesolithic[38]. In both plots, the blue, red and green thick lines and the associated lighter-color shaded areas represent respectively the running mean values and the corresponding 95.4% uncertainties, whereas the grey shaded areas represent the range of breakpoint values considered

variability of long-term demographic patterns of prehistoric foragers (iii) the population dynamics of the Late Glacial in Europe as inferred from aDNA evidence.

To explore the impact of climate and environmental change on the long-term demographic dynamics at an Iberian scale we have performed Spearman's rank correlation test between climate and palaeoenvironmental proxies (temperature, precipitation and temperate Mediterranean forests) and bootstrapped demographic SPD curves (Supplementary Discussion). The correlation coefficients and associated $p$ values as well as the variance in environmental conditions have been calculated for each demographic phase derived from the best fitting model. Because of the broad extent and biogeographic variability of Iberia, we leave for future investigation the analysis of regional population trajectories using environmentally-mediated growth models[2], to constrain the extent and intensity of demographic responses to changes in carrying capacity at sub-regional level.

During demographic phase 1, the population grew at an exponential rate. The growth rates documented during this phase fall within the range of variability of the c. 0.04% growth reported for the Great Basin in North America (13 to 6 kya), as well as for the Western Mediterranean Mesolithic[2,38]. Such a growth rate infers that these populations could have doubled in size approximately every 1700 years by virtue of intrinsic demographic growth[17]. We found population during demographic phase 1 is positively correlated with changes in temperature globally ($r = 0.53 \pm 0.14$, $p$ value < 0.001) and regionally ($r = 0.55 \pm 0.12$, $p$ value < 0.001) as well as with increases in precipitation and temperate Mediterranean forests, suggesting that population

grew steadily as climatic conditions were improving at the end of the last glacial cycle. In this context, during phase 1 foraging strategies were highly specialized in certain medium-sized ungulates (red deer and Iberian ibex) and small prey, particularly rabbit in the Mediterranean region[48–50], with a regionally variable contribution of littoral resources[51,52].

It is interesting that similar growth rates can be observed for different, seemingly unconnected populations of prehistoric hunter gatherers. This finding rehearses the recent discovery[53] that the radiocarbon records of unconnected Early Holocene societies in Europe and North America can be correlated, perhaps due to climate teleconnections or other underlying mechanisms not yet fully understood. At a general level, our results and those of other emerging studies illustrate that analyses of the radiocarbon record reveals certain recurring patterns. By extension, over the long term, the dynamics of human populations seem to have been influenced by global forces and the immutable constraints of the environment as well as local circumstances, cultural histories, and stochastic factors. In the case of hunter-gatherer growth, the similar and relatively slow growth rates shared by these various cases, which is also suggested by bioanthropological approaches[54], all point towards how human biological reproductive capacity is constrained by the realities of life shared by hunter-gatherers everywhere, likely rooted in the similar kinship structures that these societies share, and their tendency to converge upon similar simple hierarchical organizations[55].

Soon after the YD commencement (c.12.7 kya), during phase 2, we observe a significant contraction of relative population

size followed by an interval of long-sustained stationary demographic growth, covering the first half of the Early Holocene. Recent palaeoclimatic studies indicate a significant environmental impact of the YD in Iberia, with an abrupt downturn of 6 °C[56] coupled with a significant decrease in precipitation and a reduced extent of temperate Mediterranean forests[26]. After this climatic event, the first half of Early Holocene (c. 11.7-10.2 kya) witnessed a pervasive variability in atmospheric conditions, with recurring episodes of aridity[57,58]. The population proxy is weakly, if not at all, correlated with the environmental dynamics because it remained low whilst temperature and precipitation were rising and the Mediterranean forests expanding. At this scale of analysis, constant changes to the environment could in theory have imposed a carrying capacity significantly less than the circumstances prevalent at the end of Demographic Phase 1. In light of this, we suggest that the abrupt decline in population levels c. 13 kya and the long period of stagnation of phase 2 could be explained by the effect of the abrupt environmental impact of the GS-1 stadial (YD) on foraging populations who had, at the end of the GI-1 interstadial, already reached (or were close to reaching) the carrying capacity of their environment. With the current data and for the patterns documented at the Iberian scale, a dynamic forager-resource disequilibrium model[47] could provide an explanatory framework to account for the population decrease and subsequent stationary demographic growth during the first half of the Early Holocene, especially changing levels of forest cover puts a limit on available resources. In this sense the low population during times of environmental instability can be seen as an adaptation to increasingly environmentally driven changes to carrying capacity.

Finally, the best fitting model predicts a third demographic phase of logistic growth starting with the Early Mesolithic (c. 10.17 kya), with population levels stabilizing between 9.3-8 kya. This logistic regime occurred during the more stable conditions of temperature and precipitation prevalent during the second half of the Early Holocene. This is illustrated by the significantly lower variance of the paleoenvirontental proxies during this phase compared to the previous one (Supplementary Discussion). Iberian Mesolithic populations seems to have enhanced their carrying capacity broadening their diet, by increasing the exploitation of a wider range of forest-adapted medium-sized and small ungulates, and intensifying their pressure on small carnivores, littoral, and estuarine resources[59–64]. However, despite these shifts in subsistence systems from c. 9.3-8 kya, population levels did not increase steadily but rather a long-term pattern of stationary growth emerged, which seems to have been composed of a series of short-term fluctuations. It is tempting to interpret this sub-phase as a period of dynamic equilibrium, in which human population growth was continuously inhibited by carrying capacity. Aligned with this interpretation, independent evidence suggests a degree of resource depletion through increasing hunting pressure on key medium-sized ungulates throughout the Mesolithic period[63], as well as reduction on size and age distributions of marine gastropods[51] and bivalves[61].

Before we accept the proposed model of demographic dynamics, we must also discuss alternative non-demographic factors that might equally account for low frequency values in the radiocarbon SPDs during our demographic phase 2. These are sampling density bias, geologic-erosive bias, and changes in human settlement and/or mobility that pattern the archaeological record differently.

Sampling density bias can produce chronological gaps in the frequency distribution of radiocarbon dates. However, at the Iberian scale, and for most of the regional subsets, the number of dates is well above the sampling density required to avoid the presence of long gaps through stochastic processes[24,29].

Geologic-erosive bias could explain for the drop on the inferred relative population densities during phase 2 if there were widespread sedimentary hiatuses in closed sites (i.e., caves and rock-shelter deposits) on the one hand, and the destruction or deep burial of open-air sites on the other. While such a hypothesis has been recently discussed for the regional subset of the Ebro valley[22], it seems difficult to accept for the whole Iberian Peninsula, given that both cave and rock-shelter archaeological deposits were also subject to sedimentary hiatuses and stratigraphic discontinuities within phases 1[65], and 3[66], but no dramatic downturn is evident in the archaeological SPDs from these periods.

Shifts in settlement or mobility patterns could cause differing frequencies of certain site types in the archaeological record, and/or accumulation rates of anthropogenic carbon, and therefore ultimately affect radiocarbon SPDs[68]. Particularly, for the last glacial and Early Holocene European archaeological record, different studies have discussed that short-term reuse of specific locations - whether originating from recurrent seasonal patterns of residential mobility or logistic sites - could potentially result on an over-representation of these types of sites and their associated culture in the radiocarbon record. By contrast, foragers whose lifeways are less dependent on residential mobility, could be poorly represented in the radiocarbon record, as due to the way these sites are investigated, their long-term residential camps contribute relatively few dates to the overall data set[67–69]. Such scenarios deserve further investigation; one potential way forward could be the comparison of site sedimentation rates and the compactness of cultural layers between sites dating to phase 1 and those of phase 3, thereby disentangling factors of site formation, longevity, and taphonomy.

However, for the purposes of the current work, the decrease of summed probability density during phase 2 cannot be easily explained by shifts in settlement type. During phases 1 and 2, caves and rock shelters clearly dominate the record, with open-air sites not contributing much to the overall SPD (Supplementary Methods). Evidence that settlement patterns did not change much between phases 2 and 3 can be found in the number of dated open-air sites per millennia (Supplementary Methods, Supplementary Fig. 4), which does not change significantly. Two additional arguments can be made that downplay the effects of changes to mobility patterns in the reconstructed demographic trends. On one hand, the analyzed radiocarbon record was produced by non-sedentary populations of hunter- gatherers whose foraging systems mainly relied on the same core game, red deer and Iberian ibex, from the LGM to the mid Holocene[50], despite small differences in the exploitation of small prey and littoral resources. This continuity suggests that any changes to foraging mobility systems through the millennia were driven by prey density and diet broadening. We therefore maintain that population, rather than settlement pattern, is the main driver of the signals in the SPD.

The long-term demographic patterns reconstructed for Iberia provide an independent model, based on archaeological radiocarbon evidence, for contextualising the Late Glacial and Early Holocene population history of Europe and Iberia reconstructed through studies of ancient DNA (aDNA) data[19,70–72]. These studies have suggested (i) a major population turnover at the end of the Late Glacial (c. 14.5 kya), with the reduction of post-LGM haplogroups (U2'3'4'7'8'9) and the subsequent major spread of haplogroup U5;[70] (ii) a major population replacement of the El Mirón aDNA genetic cluster (dated c.19-15 kya) by the so-called Villabruna cluster (dated c.14-7 kya) in Western Europe;[19] and (iii) the admixture and persistence of populations with different relatedness with both genetic clusters in Iberia[71,72]. In Western

Europe, the 'El Mirón Cluster' consists of seven post-Last Glacial Maximum individuals from 19–14 kya, all associated with the Magdalenian culture. The Villabruna Cluster is composed of 15 post-Last Glacial Maximum individuals from across Europe dating between 14–7 kya years BP and associated with the Azilian, Epipaleolithic and Mesolithic cultures[19].

Our results are consistent with the hypothesis of a major population downturn at the end of the Late Glacial in Europe based on aDNA data[70], which predicts a population turnover around 14.5 kya. Our findings, however, suggest that in Iberia a population bottleneck -a drastic reduction in population size- started slightly later, between the end of the GIS-1 and the onset of the YD (c.13.5-12.7 kya). Such a chronological difference in the downturn of population size could be an artefact of differences on sampling density and geographic scales between aDNA and radiocarbon paleodemographic approaches. Our results also underscore the demonstrably influence of the YD on human populations in Iberia. While previous studies had detected a downturn in the SPD population proxy, especially in the north of the subcontinent[13,14], the much larger data set analysed by the current work, and the more sophisticated modelling approach, can be used to define that this downturn was rapid in nature and led to a lengthy period of stagnation. Unlike the situation during the Last Glacial Maximum[13–15], Iberia did not seem to have become a refugium during the YD. In a broader context, an increasing body of evidence shows a varying degree of impact of the YD on human populations at different Eurasian regions. In south-central Anatolia and the northern Levant, Roberts et al.[39] have detected near-hiatus levels during the YD in the Anatolian region and strongly contrasting population trajectories in the Levant, coinciding with the development of Neolithic agriculture. The link between cold YD conditions and reducing human populations has also been made for North America[73] and indeed elsewhere in Europe, where significant hydrological and ecological changes associated with the worsening climate strongly limited the extent of human settlement until the start of the Holocene[68,74].

Regarding the genetic replacement, it should be stressed that the date of the shift previously estimated in aDNA studies, 14 kya, derived from the date of the last known member of the El Mirón group, in Germany, and the first known member of the Villabruna group, a sample from Italy[19]. The paleogenomic evidence does not necessarily imply that the transition between the two genetic groups happened suddenly and simultaneously across Europe, and therefore must be addressed on a regional basis. In as far as the Iberian Peninsula is concerned, recent aDNA studies point to a different degree of admixture between these two modeled genetic groups throughout the Late Glacial and Early Holocene, forming a cline between the Late Upper Palaeolithic and the end of the Mesolithic[71]. At a sub-regional level, the situation is more complex. On the one hand, individuals with significant admixture proportions with the El Mirón cluster have been reported in Chan (NE Spain) at 9.1 kya, Carigüela (South Spain) at c.11.7–7.5 kya, Moita do Sebastião (Central Portugal) at 8.1 kya and Cingle del Mas Nou and Cocina Cave (E Spain) at 7.8 and 8 kya respectively[71,72]. On the other hand, the first significant contribution of the Villabruna lineage is found at the Balma de Guilanyà (NE Iberia) at c. 12kya, and this group seemingly becomes dominant during the Late and Final Mesolithic in the Cantabrian region at La Braña c. 7.9 kya and Los Canes c. 7.1 kya[71]. These new findings suggest regionally variable impacts of gene flux and admixture events on the genetic structure of Iberian prehistoric hunter-gatherers, whose demographic significance in terms of migration rates requires future investigation. With the regional patterns documented in our study, the increasing population trend found in the Cantabrian subset from 8.5 kya

onwards could be partially due to by the immigration of people associated with the Villabruna cluster.

In sum, this study reports the first reconstruction of long-term changes in population size during the Last Glacial-Early Holocene transition in Iberia. We have presented evidence of relative population changes and developed a best-fitting demographic model composed by three different phases. Thus, the population of Iberia increased during most of GI-1 until a rapid decrease occurred at the onset of the YD stadial, subject to certain variability at regional and sub-regional scales, which was followed by a sustained period of stationary growth. During the second half of the Early Holocene (c. 10.2-8 kya), we identified a recovery of relative population levels, when growth rates were similar or slightly greater than those of the first phase, but this soon attenuated towards a pattern of fluctuation around stationary growth. This pattern of population dynamics is in agreement with recent aDNA studies, suggesting that a major population turnover occurred in Europe at the end of the Late Glacial, but we can now also suggest that the timing of this process can be pushed slightly forward to encompass the Younger Dryas and the Early Holocene of the Iberian Peninsula and the rapid environmental changes that occurred. Our modelling results illustrate that human populations have an inherent capacity for rapid growth, but it seems that in the past this was often checked by the constraints of the environment, especially for prehistoric hunter-gatherers during episodes of climate change.

## Methods

**Database, data quality control and phasing protocols**. A new geo-referenced database of radiocarbon dates, representing the most extensive and updated data set currently available from the Iberian Peninsula for the time range under analysis, was compiled (Data Availability and Supplementary Methods). We have used 1198 radiocarbon determinations from 246 archaeological sites and 795 archaeological assemblages dated between the Early Magdalenian and the Late Mesolithic. Only radiocarbon determinations with standard errors below 200 years have been considered for this study (Supplementary Methods, Supplementary Figure 1 and Supplementary Figure 2). Most of the radiocarbon determinations come from individual samples, both short-lived materials (seeds, fruits, bone collagen and marine shells) and individual long-lived materials such as detrital and featured associated charcoal (Supplementary Methods and Supplementary Table 1). Multiple sample aggregates of charcoal and bone collagen have also been considered as valid if their uncalibrated radiocarbon chronology was consistent with their stratigraphic position and the chrono-cultural (artefact-based) interpretation of the assemblage at regional level. By contrast, land snail shell carbonates as well as bulk samples of sediment, pollen and organic pigment have been excluded from the analysis. After applying theses filtering criteria (Supplementary Table 2 and Supplementary Table 3), 291 dates were discarded (Supplementary Figure 2). Absolute radiocarbon dates have been grouped in bins of 200 years on a site basis, prior to calibration (Supplementary Methods and Supplementary Figure 5). To avoid the over-representation of dates from the same archaeological context into the same bin, we have used the R_Combine and Combine functions from OxCal, that combine statistically equivalent dates (see OxCal 4.3 Manual; further details in Supplementary Methods). After these filtering and combining processes, a total of 783 radiocarbon dates have been retained for analysis.

**Calibration procedures**. Radiocarbon dates have been calibrated using the IntCal13 atmospheric calibration curve[25] (S4). Marine shells have been calibrated using the Marine13 calibration curve and those from human bone samples with $\delta^{13}C$ values indicative of a significant marine protein intake have been calibrated using a mixed marine-terrestrial calibration curve, computing the percentage of marine diet and using local ΔR values (Supplementary Fig. 3 and Supplementary Table 4). The calibration routines were conducted using the IOSACal Python 3 library (http://iosacal.readthedocs.io/en/latest/), modified with new functionalities and coding that was developed specifically for this study (Supplementary Methods).

**Paleo-demographic modelling**. Calibrated and normalized radiocarbon dates have been aggregated in summed probability distributions for the whole Iberian Peninsula (Supplementary Methods) and for five different regional subsets. The dates were normalized, as it is a prerequisite of the bootstrapping technique (described below) that each date has equal weight in the SPD. For the purposes of

comparison, we have also calculated a SPD of unnormalized dates (Supplementary Figure 6).

SPDs were generated for open-air and cave and rockshelter sites separately, with open-air SPDs corrected for taphonomic bias[10] following the Supplementary Equation 1. The SPD was also corrected using the non-linear scaling adjustment between energy consumption and population size (Supplementary Equation 2 and Supplementary Figure 7 recently proposed by Freeman and colleagues[11]. In the context of the current study, neither adjustment is particularly significant and therefore the Freeman correction was not used in the modelling exercises.

The final SPD was a sum of these three components. To reduce the effects of [14]C laboratory errors and the spurious peaks and troughs introduced by the radiocarbon calibration process[12,75], the corrected and normalized SPDs were subjected to non-parametric bootstrapping, similarly to recent works[6,8,17]. This consists on the resampling of the original SPD curve, back-calibration and permutation of the [14]C laboratory errors[76]. The median and confidence interval of the bootstrapped samples were calculated and the result plotted against the original SPD curve and taken as our population proxy in this study (Fig. 2 and Supplementary Figure 5). The comparison between the initial radiocarbon compilation and the bootstrapped median allows one to assess whether the fluctuations can be due to the effects of the calibration process[12,75]. The comparison between the number of bootstrap simulations (Supplementary Figure 8) and the detailed analysis of the performance of the bootstrapping process (Supplementary Figure 9) demonstrates that 1000 iterations is sufficient and does not change the results. To ensure that the bootstrapping process used in this paper is robust against the choice of calibration curve used in the simulations, we have repeated the steps outlined above using different calibration curves for simulation. We have conducted the analysis using the atmospheric and marine Intcal13 curves, and by randomly allocating a specified proportion of the simulated dates to the marine [14]C axis when back-calibrating and re- calibrating. The results (Supplementary Figure 11) demonstrate that the choice of calibration curve for the simulations has no significant influence over the results.

Following recent studies[6,8,9,17], we tested the statistical significance of the changes on relative population size by fitting simulated set of SPDs describing an exponential curve to our population proxy for the entire time period of interest. This fit is considered as a model for the null hypothesis of exponential demographic growth, enveloped by the 95.4% CI from which to identify statistically significant positive and negative deviations in the reconstructed population trends. To check that the calibration curve did not influence the results of hypothesis test, we repeated this exercise using the marine calibration curve (Supplementary Figure 12).

**Correlation with palaeoenvironmental proxies**. To measure the strength of association between the Iberian population model and proxy indicators of environmental and climate change, we use Spearman's rank correlation test, applying it to each bootstrap-simulated SPD in order to express the variability in the correlation coefficients caused by errors inherent to the process of radiocarbon measurement and calibration. The proxies considered (Supplementary Discussion, Supplementary Figure 15, Supplementary Table 7) are regional and global temperature, precipitation and vegetation[26,56,77]. To derive a measure of environmental instability from the proxy data, we use linear regression to remove long-term trends from the data, then calculate the variance of the residuals and use this as an indicator of short-term variability (Supplementary Table 8).

**Model selection and demographic growth rates**. As the significance test resulted in a very small p value (Fig. 2d), we aimed to improve on the explanatory power of the null hypothesis by considering alternative, more complex, models of demographic growth. We considered five other models that introduce additional demographic phases and different curves (exponential, logistic; see Supplementary Methods). To directly compare their explanatory power, we used an Information-Theoretic based model selection approach[78]. Each model's fit to the population proxy was computed and compared, but to each was subtracted a penalty based on the models' number of parameters, i.e., more complex models were more penalized since we are looking for a parsimonious solution. We have ranked models using both Akaike's Information Criteria (AIC) (Supplementary Equations 3 and 4) and Schwarz Bayesian Criteria (SBC, also known as BIC)[35] and found no difference in ranking using these two different penalty criteria (Supplementary Table 6). Finally, the mean annual growth rates for each of the best-fitting model's phases (Supplementary Equations 5 and 6), as well as the range of its uncertainty due to radiocarbon resolution, were calculated (Table 1).

## Data availability
The data sets generated and analysed during the current study are available in the Zenodo repository, https://doi.org/10.5281/zenodo.1145698.

## Code availability
All Python and R scripts that support the findings of this study have been deposited in GitHub (https://github.com/PALEODEM/Palaeo-demographic-models) with the identifier "10.5281/zenodo.2558322" and are publicly available.

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

## Acknowledgements

This research was funded by the European Research Council Consolidator Grant-2015 number 683018 to JFL for the PALEODEM Project under the European Union's Horizon

2020 research and innovation program. FS is supported by a Marie Curie IF-2014 Fellowship number 656264. MGR was supported by the research *project MULTISCALAR-DEM* (Ref. HAR2015-70685-ERC), SL is supported by the Ramón y Cajal research program through the grant RYC-2012-01043 and JFL by the grant IEDI-2017-00889 funded by MINECO (Spain).

## Author contributions

J.F.LdP, M.G-R, F.S., R.M. and S.L. designed the research, M.G.-P. and J.F.LdP collected the radiocarbon data, M.G-R., F.S. and R.M. produced the custom computer code for the analysis and modelling of radiocarbon and paleoclimatic time series, M.G.-P., F.S., M.G-R, R.M., J.F.LdP and S.L. analyzed the radiocarbon data, J.F.LdP, R.M., F.S. and M.G.-P. wrote the manuscript with input from all authors.

## Additional information

**Competing interests:** The authors declare no competing interests.

