## [Peer Review File · Nature Communications]

Reviewer #1 (Remarks to the Author):

Review of Fernandez-Lopez de Pablo et al., Nature

There is a lot of good in this paper, and it falls within a relatively new, and exciting strain of research in archaeology as large datasets of radiocarbon dates, appropriately analyzed, provide us with a way to examine changes in population size/density for the first time at a known level of probability.

However, in its current form, the research problem and results are not significant enough for Nature. IF the authors started with the research problem of i) global population bottlenecks during the Late Glacial and human colonization processes, or ii) the problem of reconciling DNA data with radiocarbon records, they would be taking on a more significant problem in global science, rather than solely focused on a particular region of Europe. The authors would need to show why Iberia is central to global discussions of bottlenecks, or problems of ancient DNA and settling into Late Glacial/Early Holocene environments in Europe.

On the latter note, the authors are not citing some key literature on the question of the LGM-Early Holocene and questions of population bottlenecks, such as Gamble et al., *Philosophical Transactions Royal Society B*, 2004, or Tallavaara et al. *PNAS* 2015. Both papers use SPDs to focus on population dynamics from this time period. The Gamble paper is actually one of the earliest applications of 'dates as data' approaches.

The paper is too long with too many figures in the text for a Nature paper.

Substantively, the paper has various shortcomings that would require much more explanation. I mention the major ones only since I recommend that the manuscript be thoroughly reworked:

- The authors never discuss why they chose their specific null models in the first place. There is no a priori justification given as to why certain models are more relevant to discussions of population bottlenecks than others. Why, considering that the Late Glacial was the most climatically dynamic period over the past 30,000 years, should we expect an exponential null model and not a logistic null model?
- The authors begin by stating that SPDs have overcome various biases recently and are now a more secure measure of population size. However, a recent paper by Freeman et al. 2018 in *Radiocarbon* highlights the lack of a one-to-one radiocarbon date to population size ratio, and the need for a taphonomic correction is an on-going debate. Models provided here should examine what happens with and without a taphonomic correction (or justification should be made for one approach rather than another), or with/without the Freeman correction.
- The last sentence of paragraph 3 and the first sentence of paragraph 4 are good examples of how the problem would need to be framed in order to ask bigger questions that make the results significant enough for Nature. While the problem is stated in these paragraphs, it is not stated from the very first paragraph, and therefore doesn't frame the paper as well as it could.
- The bottom paragraph on page 3 highlights an interesting issue of dynamic growth models and model selection approaches. This is indeed novel, but, as mentioned above, model selection is a different matter than model choice, and therefore requires more discussion about why the specific null models were chosen in the first place.
- There is little discussion about why the particular time-period was chosen to avoid edge effects. It would be nice to see more discussion about how calibration curve effects determine this particular time window of focus. Like the null models, the authors should provide clear rationale for why they used the time-slice they did.
- The authors do not cite other papers that use similar bootstrap methods. Overall, the methods require more citations of other approaches in the literature.
- When I look at the significant spikes and troughs, I wonder how much this can be affected by choice of how many bootstrap simulations to run. Would decreasing to 1000 iterations widen the Confidence Interval to where these spikes and troughs are no longer significant?
- The authors report a decline of population around the Younger Dryas for Iberia. They find a similar result for Cantabria. A recent paper by Straus 2018 in *QI* claims that the YD had little impact on populations in this region. Importantly, Straus does not use an SPD approach; whenever possible, independent corroboration of SPDs should be attempted, as other proxies

could shed more complex light on the situation.

- The authors make a lot of claims about the relative impacts of climate events on populations, but these are just qualitative eye-balling of the data, rather than making actual statistical comparisons of archaeological with different paleoenvironmental time-series. This is not rigorous enough for a Nature publication.
- I do not understand the comment at the bottom of page 8 on smaller sample sizes being more problematic for overcoming research bias. This seems like a false representation, as certain regions can have large sample sizes of thousands of dates, but still be biased in how they were collected.
- The 'information-theoretic based model selection approach' seems new and novel, and quite useful, however, there is little explanation in the main text of how it works. If the authors are introducing a new approach, they must provide references so readers can seek out these new methods for themselves.
- As mentioned above, the authors never discuss their null model choices, and the discussion of model selection is too brief and requires more explanation to the reader.
- The authors report actual population growth rates, but as mentioned above, they should read the recent paper by Freeman et al. 2018 on how this is more challenging than previously thought.
- The authors state similar growth rates to those of Wyoming, Colorado, and the Western Mediterranean Mesolithic, but they never explain why these similar rates have any significant meaning at all. There is potentially a bigger interpretation here, but the authors do not go down this route, and it is therefore confusing to the reader as to why this matters.
- The authors use taphonomic correction for both cave and open-air sites. As detailed in the Survoell et al. 2009 paper, and applied to the Kelly et al. PNAS 2013 data, taphonomic correction does not need to be used for cave sites. This worries me as it misrepresents what the taphonomic correction should be used for, and the basic reasons for why it was developed.
- On page 15, paragraph 4, the authors mention that shifts in settlement or mobility could cause differing trends, but they never cite previous literature on this issue (Naudinot 2014; Crombe and Robinson 2014; Torfing 2016—the Naudinot paper is actually about the period of focus too).
- The first sentence of paragraph 1 on page 17 states that "Our results are consistent with the hypothesis of a major population downturn at the end of the Late Glacial in Europe". This result could be made to seem more significant if the authors had references to the wider literature on this subject. As stands, it is not for Europe, but solely for Iberia without more references.
- Why do the authors choose to bin dates from the same site that are within 200 years uncalibrated from each other? Calibrated dates for the Late Glacial often have large errors that fall outside a 200 year window. As with the null model choice, and the choice of time-slice to overcome edge effects, the authors need to provide clear rationale for why they chose to bin at 200 years.

All of the above matters because the significance of the manuscript is given in the abstract: "However, our results indicate a slightly more recent chronology providing a new demographic context to interpret the major shift of genetic groups in South Western Europe during the Last Glacial-Interglacial transition." But without attention to the various factors that can affect summed probability distributions we don't know when the decline happened (I'm also uncertain of the basis of the chronology based on genetics – that also needs to be made clear.)

Reviewer #2 (Remarks to the Author):

This paper uses a newly created and evaluated radiocarbon dataset to make inferences about changing demographic patterns in Iberia at the late Pleistocene-Holocene transition. The resulting reconstruction is then used as the basis for comparing different models of population growth over the period using AIC methods, leading to the conclusion that a 3-phase model of two exponential growth phases separated by a phase of exponential decline and stability gives by far the best fit. The results are shown to fit to inferences of population replacement that have been previously made on the basis of aDNA data.

The paper will be of great interest to a broad community of archaeologists and others in palaeostudies. The conclusions are novel and significant and the methodology sets a new standard

for this kind of work, from the impeccable care taken with the construction of the radiocarbon dataset via the construction of the population proxy to the model-testing approach to characterising the changing population growth rates. All the various steps are clearly documented and justified in the SI and the code and data are being made available, so the work is entirely reproducible.

The authors discuss and present the different results obtained depending on whether normalised or non-normalised dates are used in constructing the radiocarbon SPD population proxy but only use the normalised version as the basis for the subsequent modelling, rightly pointing out that they are similar but suggesting the need for exploration of the differences in the future. However, eyeballing of the unnormalised pattern does suggest that it would give different results in terms of the best-fit growth models and the authors might want to explore this

Response to Referees letterManuscript **NCOMMS-18-20586A****Palaeodemographic modelling supports a population bottleneck during the Pleistocene-Holocene transition in Iberia****Note to both Referees**

We would like to thank the referees for their thorough review work and constructive criticism which has considerably help us to improve the manuscript. The submitted work has been subject of a substantial revision addressing the shortcomings and suggestions raised by the reviewers. During the thorough revision and debugging process, we found a minor error in one of the Python scripts used to calibrate radiocarbon dates of marine samples, which has produced slightly different results on the SPD analysis at the end of the time series. This technical problem has been fixed, and all the statistical analyses for the paper were rerun again using revised Python and R scripts supporting the work, and new corrected versions of these have been replaced in the Github repository associated to this submission.

While this technical problem did not impact on the major findings of these work - the population bottleneck hypotheses during the Pleistocene-Holocene transition - it has certainly affected the model that best fits our demographic phase 3, which is now logistic rather than exponential. Therefore, new text has been written for describing the results for demographic phase 3, and the Discussion section has also been revised accordingly.

Further analyses have also been included in the supplementary materials that address the concerns expressed by the reviewers. In addition, as a result of this extensive review work, we have added 6 new figures in the Supplementary Information and 6 new R scripts to the Github repository.

In the following, we reply the reviewers' comments in a point-by-point manner describing the changes and corrective actions undertaken

Reviewer #1 (Remarks to the Author)

Comment 1: There is a lot of good in this paper, and it falls within a relatively new, and exciting strain of research in archaeology as large datasets of radiocarbon dates, appropriately analyzed, provide us with a way to examine changes in population size/density for the first time at a known level of probability. However, in its current form, the research problem and results are not significant enough for Nature.

Response 1: We thank the reviewer for his/her positive appreciation about the potential scientific interest of our work. We would like to clarify that our manuscript was submitted for *Nature Communications* and not to *Nature*, so we believe that the number of figures and extent are appropriate for the targeted journal.

Following the reviewer#1 suggestions, we have made more explicit the research problem and the significance of the results by modifying the structure of the Introduction and Discussion sections.

Comment 2: IF the authors started with the research problem of i) global population bottlenecks during the Late Glacial and human colonization processes, or ii) the problem of reconciling DNA data with radiocarbon records, they would be taking on a more significant problem in global science, rather than solely focused on a particular region of Europe. The authors would need to show why Iberia is central to global discussions of bottlenecks, or problems of ancient DNA and settling into Late Glacial/Early Holocene environments in Europe.

Response 2: We thank the reviewer for his/ her suggestions. The introductory section has been modified to better frame the research problems discussed of this paper. Particularly, we have clearly stated why Iberia is central to global discussions of bottlenecks and the reconciliation of aDNA and archaeological data by adding the following text:

“The Iberian Peninsula, because of its cul-de-sac position in the southwestern extreme of the Eurasian continent, remains central to discussions about demographic processes involving human refuges, migrations, population admixture and replacements. In as far as the Last Glacial-Early Holocene transition is concerned, Iberia provides a unique opportunity for reconciling population histories drawn from ancient DNA (aDNA) and radiocarbon paleodemographic modelling, a major methodological challenge in population paleo studies (20)”.

Comment 3: On the latter note, the authors are not citing some key literature on the question of the LGM-Early Holocene and questions of population bottlenecks, such as Gamble et al., *Philosophical Transactions Royal Society B*, 2004, or Tallavaara et al. *PNAS* 2015. Both papers use SPDs to focus on population dynamics from this time period. The Gamble paper is actually one of the earliest applications of ‘dates as data’ approaches.

Response 3: Both references have been included in the revised version of the manuscript along the following text:

“Following the SPD methodology, previous studies have shown a causal relationship between climatic deterioration and population contraction during the Last Glacial Maximum (LGM c.23-19 kya) in Europe (13, 14), a finding independently validated through climate envelope modeling (15)”.

Comment 4: Substantively, the paper has various shortcomings that would require much more explanation. I mention the major ones only since I recommend that the manuscript be thoroughly reworked.

Response 4: we address point by point (Comments & Responses 5-23) the various shortcomings identified by the reviewer along with the detailed responses and the corrective actions undertaken in the revised manuscript.

Comment 5: The authors never discuss why they chose their specific null models in the first place. There is no a priori justification given as to why certain models are more relevant to discussions of population bottlenecks than others. Why, considering that the Late Glacial was the most climatically dynamic period over the past 30,000 years, should we expect an exponential null model and not a logistic null model?

Response 5: We decided to test the exponential growth as a null hypothesis because this model is the most suitable for exploring long-term (millennial-scale) changes in population size. Previous work has demonstrated that long-term demographic regimes of exponential growth are a reasonable fit for the data in case studies based on the analysis of the radiocarbon record of prehistoric hunter-gatherers (e.g. Zahid et al., 2016). As we explain in the paper, testing the null model of exponential growth with our empirical SPD, and identifying negative or positive departures from the null model is a reasonable first analytical step before exploring more complex models. Furthermore, the exponential growth model better fulfils the expectations of long-term human population dynamics, compared to for example a uniform model of demographic growth, whose heuristic power remains to be demonstrated for millennial timescales.

Following the reviewer suggestion, to make more explicit the choice of specific null models we have added the following text (Results - Relative population changes): *“The exponential model was chosen because it is the only simple function that provides an adequate null model of millennial-scale changes in population size. Of the alternatives, a uniform null model implies a regime of stationary growth arising from complex interactions of fertility and migration and is thus not appropriate as a null model for such a long chronological span. On the other hand a logistic model implies a carrying capacity threshold, which in any case could change with time, especially in the context of the present study, which is conducted over the dynamic environments of the Pleistocene-Holocene transition. Hence the exponential null model was used to This fit is considered as a null model of exponential demographic growth (see Materials and Methods and S6 for more details) The null model is used from which to identify positive and negative deviations in the reconstructed population trends (see Materials and Methods and S6 for more details)”*

The reviewer makes the point that a logistic null model should be tested as well, considering the dramatic climatic changes of the Late Glacial period (indeed, to this we would also add the Last Glacial-Interglacial transition). We thank the reviewer for this suggestion. Consequently, we have added and tested a single phase logistic null model (Model B) for the time range under analysis (Results-Growth rates).

Comment 6: The authors begin by stating that SPDs have overcome various biases recently and are now a more secure measure of population size. However, a recent paper by Freeman et al. 2018 in Radiocarbon highlights

the lack of a one-to-one radiocarbon date to population size ratio, and the need for a taphonomic correction is an on-going debate. Models provided here should examine what happens with and without a taphonomic correction (or justification should be made for one approach rather than another), or with/without the Freeman correction.

Response 6: we thank the reviewer for his/her suggestion of applying and discussing the recently published Freeman et al. 2018 correction in our case study. We have included in our introduction to the paper a reference to the Freeman et al. 2018a paper published in *Radiocarbon* and also a more recent paper from the same author and colleagues (Freeman et al.2018b) published in *PNAS* in October (after we received the reviewers' comments). In addition, in the *Supplementary Information* (S6.1, Figure S7) we have added a new section with additional text and one plot to examine in detail the differences in the SPD without taphonomic correction, with the Freeman correction and the Surovell correction. Finally, the r script used to run the Freeman correction has also been included into the Github repository linked to this manuscript. In the context of the current study, as this new section demonstrates, neither adjustment has a significant bearing over the results.

Comment 7: The last sentence of paragraph 3 and the first sentence of paragraph 4 are good examples of how the problem would need to be framed in order to ask bigger questions that make the results significant enough for Nature. While the problem is stated in these paragraphs, it is not stated from the very first paragraph, and therefore doesn't frame the paper as well as it could.

Response 7: We thank the reviewer for his/her advice. As stated in our previous replies to comments 1 and 2, we have modified the introduction section to better frame the research questions and the significance of our results.

Comment 8: The bottom paragraph on page 3 highlights an interesting issue of dynamic growth models and model selection approaches. This is indeed novel, but, as mentioned above, model selection is a different matter than model choice, and therefore requires more discussion about why the specific null models were chosen in the first place.

Response 8: we have taken this helpful suggestion on board, and expanded the text to make explicit the thinking behind our selection of models to fit to the empirical data. The rationale behind our model selection is that since the null hypothesis of exponential growth can be rejected, more complex models are necessary to explain the population dynamics of the periods in question. This allows us to frame certain hypotheses in a way that would otherwise be impossible to do empirically, and thus compare the archaeological record to models of population dynamics that follow from ecology.

We are careful, however, not to 'over-fit' our explicatory framework though the use of information criteria that balance the goodness-of-fit of the models with their

parsimony, limiting our demographic phases to breakpoints informed by our prior understanding of cultural and environmental change. Even so, we have remained aware of the chronological uncertainties of radiocarbon time series, and therefore we carefully explore different breakpoints between the demographic phases, using a large number of computer simulations.

New text has been added discussing the underlying rationale of the one-phase and two-phase models in the updated text:

“The six selected models increase in complexity -two single phase models (A and B), two two-phase models (C and B) and two three-phase models- and assume different hypotheses about demographic growth regimes for each phase and the transitions between phases. The two simplest ones (A and B) fit a simple parametric growth model -exponential and logistic- for the time range under analysis (c.16.6-8 kya). The exponential growth has been widely applied to study short-term demographic bursts of invasive species occupying new niches (36) as well as recent explosive human demographic growth (37). In recent radiocarbon palaeodemographic research, exponential growth has been empirically demonstrated for long-term demographic dynamics of prehistoric hunter-gatherers (17), while in other studies it has been proposed as a null hypothesis to infer phases of statistically significant high or low population levels (6, 8, 9, 38, 39). Under an exponential growth, our Model A assumes long-term continuous, gradual and uninhibited population increase from the beginning of the Upper Magdalenian to the end of the Late Mesolithic.

The second single-phase demographic model is the Model B, fitting a logistic function (40) in which the growth rate decreases as the density of population approaches carrying capacity, which is the population threshold that a given environment can support. This kind of density-dependent growth has been explored in different simulations of Late Pleistocene human dispersals (41, 42) and empirically tested in more recent radiocarbon paleodemographic studies on both prehistoric hunter-gatherers (18) and agriculturalists (43)”.

Especially pertinent to our study is discussion on the uniqueness of process that operate upon human populations during phases of lowered population density. These include references to demographic phenomena including Allee effects, ‘boom and bust’ cycles of population growth and decline, and the demographic consequences of foraging-resource disequilibrium.

The updated text (in section Results-Growth rates) includes “*an exponential growth is fitted to the first phase, spanning the beginning of the Upper Magdalenian to the YD commencement (c.16.6-12.9 kya), whereas the second phase (12.9-10.2 kya) is modelled as an exponential decay towards a limiting value. Importantly, it should be noted this second phase contrasts with the classic parametric growth models (e.g. exponential and logistic) used for models A-D. Instead of assuming an abrupt and catastrophic decline of population associated with the YD (as proposed in models C and D), this phase 2 aims to model a more complex transition from negative to stationary (null) demographic growth, where migration, natural fertility and death rates were all held in balance. Such an assumption can be derived from other models or demographic hypotheses about processes operating during phases of lowered population levels or densities. One*

such model is that of strong Allee effects (Courchamp et al., 1999), which in the case for humans and primates foresees negative growth rates if populations fall below a critical density threshold (Steele 2009). Other relevant hypotheses are the stationary demographic growth that followed the boom and bust pattern identified after the Neolithic demographic transition (Shennan 2013); or the foraging-resource disequilibrium model (Winterhalder and Goland 1993), which predicts stationary demographic growth under environmentally driven changes in carrying capacity”.

Also pertinent to this comment is our reply to comment 3.

Comment 9: There is little discussion about why the particular time-period was chosen to avoid edge effects. It would be nice to see more discussion about how calibration curve effects determine this particular time window of focus. Like the null models, the authors should provide clear rationale for why they used the time-slice they did.

Response 9: The *Supplementary Information* text (section S6.4) has now been clarified to explain that the time window is defined by the research agenda, and is not a methodological issue caused by calibration curve effects. As we have now explained more fully, there are two issues here. The first is common to any time-series based analysis, as we explain “*Although we collected data spanning 22,000 (from the onset of the Magdalenian culture) and 7500 cal BP (the beginning of the Early Neolithic in Iberia), we have restricted our analysis to the time period 18,000 to 8000 cal BP. In this way the analysis is not hampered by summed probability scores affected by a lack of research pressure at the beginning and end of the study period, which would cause the probability scores to fall towards zero.*”

The second issue relates to edge effects and is particular to the archaeology of the Iberian region. We have added the following text for S6.4 in the supporting information to explain “*The 8000 cal BP cut-off was also chosen to avoid ‘the edge effects’ relating to potential interactions with the beginning of the Neolithic in Iberia. This process introduced to Iberia new cultural process that were not intrinsic to the dynamics of the hunter-gatherer societies that existed beforehand, and therefore not relevant for the purposes of our study. Furthermore, the timing of the start of the Neolithic was not uniform across Iberia, as it occurred later in Atlantic and Cantabrian regions than elsewhere. By limiting our model fitting analyses to events before 8000 cal BP we avoid these complexities and ensure our study is focussed on endemic processes. This has the added advantage of excluding Mesolithic data post-dating 8000 cal BP from our analyses. These data derive from sites that are highly visible, especially in the Atlantic region, and have been intensely investigated by archaeologists and thus represent an example of positive research bias.*”

Comment 10: The authors do not cite other papers that use similar bootstrap methods. Overall, the methods require more citations of other approaches in the literature.

Response 10: Following reviewer's suggestion, we have further expanded the number of citations regarding the bootstrap techniques applied to time-series analyses. In the manuscript, at the Materials and Methods section "[...] *the corrected and normalized SPDs were subjected to non-parametric bootstrapping, similarly to recent works (6, 8, 13, 47)*". Additionally, at the Supplementary Information, Section 6.2, we have included references from other disciplines "*The application of bootstrap techniques at the radiocarbon dates time-series analyses, trace its roots back to the seminal paper of Rick (1987) "Dates as Data", and have become very popular between a broad range of other scientific fields such as economics (Khim-Sen Liew, 2008), or ecological fields (Nahorniak et al 2015). Focusing on archaeological and anthropological research, bootstrap resampling method has been recently used to show uncertainty in population density estimation (Shennan et al. 2013), to test the statistical significance of the observed patterns in radiocarbon datasets analyzed (Timpson et al. 2014; Manning and Timpson, 2014), or even to examine sampling errors in datasets (Zahid et al 2013)*". We have also developed the technical explanation of this non-parametric random resampling method applied in our case study, including a specific section about the calibration and back-calibration procedures chosen for building the simulated dataset (Supporting Information S6.6).

Comment 11: When I look at the significant spikes and troughs, I wonder how much this can be affected by choice of how many bootstrap simulations to run. Would decreasing to 1000 iterations widen the Confidence Interval to where these spikes and troughs are no longer significant?

Response 11: We have added a plot containing a comparison of 100, 1000 and 10,000 bootstrap simulations, which demonstrates that 1000 iterations is more than sufficient and do not change the significance of the spikes and troughs. Furthermore, we have now undertaken a detailed analysis of the performance of the bootstrapping process with respect to convergence (the number of bootstrap iterations needed to achieve a stable simulation), and included this in the Supporting Information (S6.3).

Comment 12: The authors report a decline of population around the Younger Dryas for Iberia. They find a similar result for Cantabria. A recent paper by Straus 2018 in QI claims that the YD had little impact on populations in this region. Importantly, Straus does not use an SPD approach; whenever possible, independent corroboration of SPDs should be attempted, as other proxies could shed more complex light on the situation.

Response 12: We would like to clarify that we did not find exactly the same results for the Cantabrian subset than in the rest of Iberia. As it is stated in the results section: "*Under the general pattern of population increase during at the end of the Last Glacial period, it is worthy to note slight inter-regional differences in the turning point of population decline. For instance, in Cantabria and the Ebro valley subsets it is dated c. 13.5 kya during the final stages of GI-1, whereas in the Mediterranean region it is dated to c. 12.9-12.7 kya, during the onset of the*

YD. Such divergences suggest different population responses to regionally-specific conditions which require further investigation using multi-proxy archaeological and palaeoenvironmental evidence at subregional level. In the Cantabrian region, it seems plausible that the high population (25) reached a natural carrying capacity at this time, tilting the trend towards long-term decline”.

We cite the Straus 2018 QI paper mentioned by the reviewer 1 in which the author concludes “*there is no clear-cut evidence that the onset of Younger Dryas episode caused a major disruption of the Final Magdalenian-Azilian cultural continuum (see Straus, 2011). The chronological resolution is still too poor and pollen and faunal records are still too sparse to detect possible subtle changes in human adaptations during the course of YD, but at a general level there seems to have been considerable continuity in human settlement (both low- and highland) and subsistence based fundamentally on red deer, ibex, smaller numbers of roe deer, chamois and boar together with marine molluscs (and some land snails) and fish*” (Straus 2018: 225).

We fully agree with the reviewer that independent multi-proxy corroboration of the SPD (eg. measures of faunal diversity such as Simpson’s Inverse diversity species Index, stable isotope analyses or artifact densities) are needed, but this is beyond the scope and extent of this manuscript (while it is considered in another manuscript under preparation, focused on the Cantabrian regional subset). We would like to add that some recent studies have found subtle differences between the Upper Magdalenian and the Azilian periods regarding faunal species diversity measures (Arroyo Marín 2014; Jones 2015) and retouch frequency/sediment volume (Clark, Barton and Straus QI, in press). However, those studies suffer of the lack of chronological resolution for assessing the impact of the YD, because they rely in broader chrono-cultural divisions such as the “Azilian” which comprises archeological evidences predating and postdating the YD.

It should be noted that the mentioned paper of Clark, Barton and Straus (QI, in press) suggests the negative impact of the YD on demography, although that work relies on a much less sophisticated SPD analysis than the one we present here.

Comment 13: The authors make a lot of claims about the relative impacts of climate events on populations, but these are just qualitative eye-balling of the data, rather than making actual statistical comparisons of archaeological with different paleoenvironmental time-series. This is not rigorous enough for a Nature publication.

Response 13: The main goal of this paper is to model long-term demographic patterns at the Iberian scale. Considering the considerable biogeographic variability of Iberia, the relationship between climate, environment and population dynamics is one which we feel really needs to be approached first at regional and sub-regional levels, and is something we must leave for future work. However, taking in to account this reviewer’s suggestion, we have now included a first exploration how global and Iberian environmental proxy data can correlate with the demographic proxy of radiocarbon. This is now included in the manuscript

and further detailed in the *Supplementary Information* (section S8). In this section, we use multiple iterations of Spearman's rank correlation test with the environmental proxies and bootstrapped demographic curves. We did this separately for the three demographic phases identified in the paper, and comment on the results. In general, there are significant correlations, especially overall and as temperatures first increased, before the Younger Dryas. To explain the lack of correlation during later demographic phases, we have also turned to studying the variability inherent in the environmental proxies, as this was greater during the second demographic phase and may have been one of the factors that limited population growth. This is all now explained in the *Supplementary Information* (end of section S8).

Comment 14: I do not understand the comment at the bottom of page 8 on smaller sample sizes being more problematic for overcoming research bias. This seems like a false representation, as certain regions can have large sample sizes of thousands of dates, but still be biased in how they were collected.

Response 14: to clarify this point let us reproduce, first, the comment at the bottom of page 8: *“Obviously, the division of the radiocarbon record into smaller regional or subregional subsets reduces the sample size and sampling density at the expense of statistical significance of the inferred trends. In turn, as discussed in S7, the smaller sample increases the effect of non-demographic confounding factors such as research and sampling biases inherent to the observed trends. For these reasons, a conservative approach was taken to the interpretation of regional patterns”*.

This comment is just a cautionary tale about the interpretation of regional patterns on virtue of their sample size. Basically, we signify that the smaller a sample is (if it is subdivided in regional, subregional or local subsets) the bigger is the probability of capturing research and sampling biases instead of actual changes in human activity, population levels or energy consumption. We agree with the reviewer 1 that in general, regions with well powered sample sizes could be, still, subject of research and sampling biases. That is one of the reasons why we decided to further explain research and sampling biases on a regional basis as it is in the supplementary materials.

Comment 15: The ‘information-theoretic based model selection approach’ seems new and novel, and quite useful, however, there is little explanation in the main text of how it works. If the authors are introducing a new approach, they must provide references so readers can seek out these new methods for themselves.

Response 15: we thank the reviewer for his/her positive appreciation about the utility of the information-theoretic based model selection approach applied in our manuscript. In benefit of the manuscript extent, we decided to focus on the results while further expanding the explanation of the model selection methodology in the supplementary material. Following the reviewer's suggestion, we have expanded the text explaining how the model selection works in the revised

manuscript, especially regarding the number of phases and the choice of breakpoints within phases assuming some degree of chronological uncertainty. *“Therefore, we have explored alternative, more complex hypotheses using an information-theoretic based model selection approach. Rather than opting for a full dynamic growth rate, as proposed by Brown (16), we opt for a more parsimonious approach that keeps the number of changes in demographic regime to a minimum, whilst remaining alive to two natural breakpoints that coincide with well-established climate and environmental transitions in Iberia. The first of these relates to the onset of the YD around 12.9 kya and the second with the onset of the Early Mesolithic around 10.2 kya”*

and also further explaining the methodology underlying the breakpoints and the measures of uncertainty

“In the model fitting process, the values of the breakpoints between phases are important parameters as they influence the overall goodness-of-fit. For the results presented in this paper, we consider two already mentioned breakpoints, corresponding to the onset of the YD (12.9 kya) and the Early Mesolithic (10.2 kya). However, the timing of these transitions is not as sharp and well-dated as these often-cited dates seem to portray them, nor do they occur contemporaneously across the whole of the Iberian Peninsula. To account for this, we have allowed these breakpoints to vary by one-year increments ranging 150 years before and after the above-mentioned dates. This provides a measure of uncertainty around our inferences, especially the growth rates, and accounts for the sub-regional variability in the timing of the transitions”.

Comment 16: As mentioned above, the authors never discuss their null model choices, and the discussion of model selection is too brief and requires more explanation to the reader.

Response 16: we think the reviewer is completely right but we find this comment has been already replied in our response to Comment 6. Therefore, we refer to the editor and the reviewer to our extended reply on Comment 6, where we explain the changes introduced in the main text to better discuss the null model choices and the assumptions of the different models compared in the model selection.

Comment 17: The authors report actual population growth rates, but as mentioned above, they should read the recent paper by Freeman et al. 2018 on how this is more challenging than previously thought.

Response 17: Section S6 in the supplementary information now explains how readers may factor in the adjustment proposed by Freeman et al 2018a if they so wish. We write *“Freeman and colleagues have proposed a different scaling factor that aims to correct SPDs for the sub-linear relationship between energy consumption and population size they have observed for modern case studies. We compare the taphonomically corrected SPD for all the dates under analysis with an uncorrected curve and one re-scaled according to Freeman et al. In the context of the current study, neither adjustment is particularly significant. The*

Freeman adjustment is a preliminary model of true relationship between human population level and the production of archaeological materials, but its universal application is a matter for debate. It may predict a sub-linear relationship between the radiocarbon record and other archaeological proxy measures of population, but to the best of our knowledge this is yet to be demonstrated empirically using archaeological data. Therefore, we have not applied it to the SPDs in the current study. Reads curious about now this adjustment affects the modelled growth rates fitted to the SPDs can revise our values by factor of 1.15.” We have also referred to the paper by Freeman and colleagues in the introduction to our paper.

Comment 18: The authors state similar growth rates to those of Wyoming, Colorado, and the Western Mediterranean Mesolithic, but they never explain why these similar rates have any significant meaning at all. There is potentially a bigger interpretation here, but the authors do not go down this route, and it is therefore confusing to the reader as to why this matters.

Response 18: This discussion was set-up in page 2 of the version of the paper originally submitted, and we accept that the discussion of the results in this regard could have been more fully developed. To address this shortcoming, we have added substantively to the discussion, including the following text:

“It is interesting that similar growth rates can be observed for different, seemingly unconnected populations of prehistoric hunter gatherers. This finding rehearses the recent discovery (Freeman et al 2018) that the radiocarbon records of unconnected Early Holocene societies in Europe and North America can be correlated, perhaps due to climate teleconnections or other underlying mechanisms not yet fully understood. At a general level, our results and those of other emerging studies illustrate that analyses of the radiocarbon record reveals certain recurring patterns. By extension, over the long term, the dynamics of human populations seem to have been influenced by global forces and the immutable constraints of the environment as well as local circumstances, cultural histories, and stochastic factors. In the case of hunter-gatherer growth, the similar and relatively slow growth rates shared by these various cases all point towards how human biological reproductive capacity is constrained by the realities of life shared by hunter-gatherers everywhere, likely rooted in the similar kinship structures that these societies share, and their tendency to converge upon similar hierarchical organizations (Hamilton et al 2007)”.

Comment 19: The authors use taphonomic correction for both cave and open-air sites. As detailed in the Surovell et al. 2009 paper, and applied to the Kelly et al. PNAS 2013 data, taphonomic correction does not need to be used for cave sites. This worries me as it misrepresents what the taphonomic correction should be used for, and the basic reasons for why it was developed.

Response 19: I am afraid that the reviewer got confused in this point. We never applied taphonomic correction for both cave and open-air sites but just on open-air sites (following Surovell et al., 2009), as it was stated in part S5 of the *Supplementary Information* of the submitted manuscript (now in section S6.1 of

the revised *Supplementary Information*). To further clarify this issue, in the methods section of the revised manuscript we have also stated how the taphonomic correction applied only to on open-air sites.

Comment 20: On page 15, paragraph 4, the authors mention that shifts in settlement or mobility could cause differing trends, but they never cite previous literature on this issue (Naudinot 2014; Crombe and Robinson 2014; Torfing 2016—the Naudinot paper is actually about the period of focus too).

Response 20: we thank to the reviewer for his/ her bibliographic suggestions on previous literature about the potential effects of mobility patterns in the SPD inferred demographic trends. We have cited those references and added the following text in the main manuscript: *“Particularly, for the last glacial and Early Holocene European archaeological record, different studies have discussed that short-term reuse of specific locations - whether originating from recurrent seasonal patterns of residential mobility systems or logistic sites - could potentially result on an overrepresentation of these types of sites and their associated culture in the radiocarbon record. By contrast, foragers whose lifeways are less dependant on residential mobility could be poorly represented in the radiocarbon record, as due to the way these sites are investigated their long-term residential camps contribute relatively few dates to the overall dataset”*

Comment 21: The first sentence of paragraph 1 on page 17 states that “Our results are consistent with the hypothesis of a major population downturn at the end of the Late Glacial in Europe”. This result could be made to seem more significant if the authors had references to the wider literature on this subject. As stands, it is not for Europe, but solely for Iberia without more references.

Response 21: We have followed up on this helpful suggestion added the following brief review of the evidence from elsewhere:

‘Other cases where the YD has demonstrably influenced human populations at a regional level include south-central Anatolia and the northern Levant, where Roberts et al (2018) have detected near-hiatus levels spanning the period in the Anatolian region that strongly contrasts with rising population in the Levant -- the trend in the latter case ultimately coinciding with the development of Neolithic agriculture. The link between cold YD conditions and reducing human populations has also been made for North America (Anderson et al 2011) and indeed elsewhere in Europe, although previous conclusions that this did not apply in the case of Iberia (Gamble et al 2006) can now be revised thanks to the larger dataset dataset analysed by the current work.’

Comment 22: Why do the authors choose to bin dates from the same site that are within 200 years uncalibrated from each other? Calibrated dates for the Late Glacial often have large errors that fall outside a 200 year window. As with the null model choice, and the choice of time-slice to overcome edge effects, the authors need to provide clear rationale for

why they chose to bin at 200 years.

Response 22: We set the bin bandwidth at 200 yrs because the mean of the laboratory errors in the study is 115 ¹⁴C years, and thus 200 years is a heuristic value for identifying dates that may derive from the same phase of activity. Any larger, and we run the risk of mistakenly combining phases of activity that were, in reality, separate in time, which could be a more serious error with regards to the results. In other words, the bandwidth was chosen to balance Type-I and Type-II errors. This has now been clarified in S3 of the *Supplementary Information*.

Comment 23: All of the above matters because the significance of the manuscript is given in the abstract: “However, our results indicate a slightly more recent chronology providing a new demographic context to interpret the major shift of genetic groups in South Western Europe during the Last Glacial-Interglacial transition.” But without attention to the various factors that can affect summed probability distributions we don’t know when the decline happened (I’m also uncertain of the basis of the chronology based on genetics – that also needs to be made clear.)

Response 23: All the modifications made to the paper discussed above make it clear how we have dealt with the uncertainties of the radiocarbon record, and how the bootstrapping and model selection procedures that allow us to frame ‘when’ the decline happened within a specified degree of uncertainty.

We have also followed-up on this helpful advice more specifically, enlarging the discussion of the aDNA evidence to make the genetic chronology and its uncertainties more explicit. We added to our paper the following text ‘*The date of the shift, 14 kya, derives from the date of the last known member of the El Mirón group, in Germany, and the first known member of the Villabruna group, a sample from Italy. The aDNA data do not necessarily imply that the transition between the two groups happened suddenly and simultaneously across Europe. In an Iberian context, all we can infer from aDNA is the transition happened between ca. 14 kya cal BP (first detection in Europe) and ca. 8 kya (first and only detection in Iberia). ... The population decline we have modelled to ca. 12.8 kya and the subsequent period when population remained low are plausible suggestions for when the transition occurred. Alternatively, the replacement could have occurred during demographic phase 3 (10.2-8 kya), in the context of increasing population levels that were increasing at an unprecedented rate, perhaps at least partially due to the immigration of people associated with the Villabruna cluster.*’

Reviewer #2 (Remarks to the Author)

Comment 1: This paper uses a newly created and evaluated radiocarbon dataset to make inferences about changing demographic patterns in Iberia at the late Pleistocene-Holocene transition. The resulting reconstruction is then used as the basis for comparing different models of population growth over the period using AIC methods, leading to the conclusion that a 3-phase

model of two exponential growth phases separated by a phase of exponential decline and stability gives by far the best fit. The results are shown to fit to inferences of population replacement that have been previously made on the basis of aDNA data.

The paper will be of great interest to a broad community of archaeologists and others in palaeostudies. The conclusions are novel and significant and the methodology sets a new standard for this kind of work, from the impeccable care taken with the construction of the radiocarbon dataset via the construction of the population proxy to the model-testing approach to characterising the changing population growth rates. All the various steps are clearly documented and justified in the SI and the code and data are being made available, so the work is entirely reproducible.

Response 1: We thank the reviewer for their positive criticism of our work.

Comment 2: The authors discuss and present the different results obtained depending on whether normalised or non-normalised dates are used in constructing the radiocarbon SPD population proxy but only use the normalised version as the basis for the subsequent modelling, rightly pointing out that they are similar but suggesting the need for exploration of the differences in the future. However, eyeballing of the unnormalised pattern does suggest that it would give different results in terms of the best-fit growth models and the authors might want to explore this.

Response 2: The method presented in this paper for bootstrapping both the population proxy and the exponential null model require normalization as an important step in the analysis. Normalisation ensures the probability densities for each time coordinate in the study are equivalent, which becomes critical when they are re-sampled for the purposes of simulation. Therefore, other than a visual comparison of normalised and unnormalised curves, further quantitative analysis of the modelling presented here is not possible using the bootstrapping methods presented here. A note explaining this has now been added to the Supporting Information (section S6):

“However, in this work, we have decided to use normalized, instead of unnormalized, probability densities because the bootstrapping process isn't contingent upon sampling from probability distributions that integrate the same probability space. Had we not normalised the results of calibration and summing, the samples drawn from the SPD would have no statistical meaning. There are two more reasons why normalised dates are preferred: Firstly, to facilitate the comparison of our results on relative population changes and growth rates with other similar SPD case-studies that equally used normalized probability densities (29-31). And secondly, because an exploratory comparison of the SPDs constructed from normalized vs. unnormalized probability densities based on the PALEODEM dataset has yielded minor differences in the observed long-term pattern (Fig. S6). We acknowledge those minor differences might deserve further statistical investigation in the future but any attempt to bootstrap confidence

intervals from SPDs must by definition rely on normalisation of the probabilities at some point in the analysis”.

Therefore, we acknowledge that the unnormalised SPD might deserve future statistical investigation in terms of the best-fit growth, but this is beyond the extent and scope of this manuscript.

Reviewer #2 (Remarks to the Author):

The authors have addressed all the reviewer comments in detail and very convincingly and have greatly improved the paper as a result. This is an excellent and important paper and I recommend acceptance.

Reviewer #3 (Remarks to the Author):

The authors have sufficiently commented on all the reviewer comments from the first draft. Their corrections and additions to the text and SI are sufficient for full publication of this manuscript. The authors should be commended in their revision work to improve the strength of this paper.

This is an important paper, and therefore deserves publication in this journal, for two reasons: 1) The authors report an important point of comparison and complementarity between radiocarbon and aDNA records; 2) the authors develop a new approach for evaluating different population growth models from empirical radiocarbon time series.

My only complaint is that, when I went to GitHub to search for the data and code, I was unable to find them. I would have liked to see these before the submission of the manuscript for review. While this is not detrimental for me to suggest publication, I do insist that the editor requires these data and code to be published before the manuscript has been sent to press.

Manuscript NCOMMS-18-20586B**Fernández-López de Pablo et al. Palaeo-demographic modelling supports a population bottleneck during the Pleistocene-Holocene transition in Iberia****Final Response to Referees letter**

We would like to thank the referees for their positive comments on the revised version (NCOMMS-18-20586A) of the manuscript. Following the Nature Communication guidelines, we provide a point by point response to the referees' comments.

Reviewer #2 (Remarks to the Author)

Comment 1: The authors have addressed all the reviewer comments in detail and very convincingly and have greatly improved the paper as a result. This is an excellent and important paper and I recommend acceptance.

Response 1: We thank the reviewer #2 for his/her positive appreciation about our revision and scientific interest on our work.

Reviewer #3 (Remarks to the Author)

Comment 1: The authors have sufficiently commented on all the reviewer comments from the first draft. Their corrections and additions to the text and SI are sufficient for full publication of this manuscript. The authors should be commended in their revision work to improve the strength of this paper.

Response 1: We thank the reviewer #3 for his/her positive appreciation about our revision work. We feel his/her comments on the previous version of the manuscript has helped us to improve the strength of this paper.

Comment 2: This is an important paper, and therefore deserves publication in this journal, for two reasons: 1) The authors report an important point of comparison and complementarity between radiocarbon and aDNA records; 2) the authors develop a new approach for evaluating different population growth models from empirical radiocarbon time series.

Response 2: We thank the reviewer #3 for highlighting the scientific relevance of our study.

Comment 3: My only complaint is that, when I went to GitHub to search for the data and code, I was unable to find them. I would have liked to see these before the submission of the manuscript for review. While this is not detrimental for me to suggest publication, I do insist that the editor requires these data and code to be published before the manuscript has been sent to press.

Response 2: We thought that both the code and the data were available when we first submitted the manuscript and the revised version. We apologize for not doing it before. As requested by the reviewer #3, we have just made public the data and codes produced for this paper.

- GitHub (Programming codes): <https://github.com/PALEODEM/Palaeo-demographic-models>
- Zenodo (Data): <https://doi.org/10.5281/zenodo.1145698>